# RoRA-VLM: Robust Retrieval Augmentation for Vision Language Models

## Abstract

Though vision-language models (VLMs) have demonstrated impressive capabilities as general-purpose visual assistants, they still exhibit inferior performance on knowledge-intensive tasks such as information-seeking visual question answering, primarily due to the challenge of accurately encoding all the associations between visual objects and scenes to their corresponding entities and background knowledge. While retrieval augmentation methods offer an efficient way to integrate external knowledge, extending them to vision-language domain presents unique challenges in (1) precisely retrieving relevant information from external sources due to the inherent discrepancy within the multimodal queries, and (2) being resilient to the irrelevant, extraneous and noisy information contained in the retrieved multimodal knowledge snippets. In this work, we introduce RoRA-VLM, a novel and robust retrieval augmentation framework specifically tailored for VLMs, with two key innovations: (1) a 2-stage retrieval process with Image-anchored Textual-query Expansion to synergistically combine the visual and textual information in the query and retrieve the most relevant multimodal knowledge snippets; and (2) a robust retrieval augmentation method that strengthens the resilience of VLMs against irrelevant information in the retrieved multimodal knowledge by injecting adversarial noises into the retrieval-augmented training process, and filters out extraneous visual information, such as unrelated entities presented in images, via a query-oriented visual token refinement strategy. We conduct extensive experiments to validate the effectiveness and robustness of our proposed methods on three widely adopted benchmark datasets: OVEN, InfoSeek and Enc-VQA. Our results demonstrate that with a minimal amount of training instance, RoRA-VLM enables the LLaVA-v1.5 model to achieve significant performance improvement and constantly outperform state-of-the-art retrieval-augmented VLMs on all benchmarks while also exhibiting a novel zero-shot domain transfer capability.

## 1 Introduction

Vision-language models (VLMs) (Li et al., 2023; Alayrac et al., 2022; Liu et al., 2023b; Dai et al., 2023), built on pre-trained visual encoders and large language models (LLMs), have achieved remarkable progress across a range of visual perception and generation tasks (Antol et al., 2015; Marino et al., 2019; Dai et al., 2024). However, despite these advancements, recent studies (Chen et al., 2023d; Hu et al., 2023; Mensink et al., 2023) reveal that VLMs still face significant challenges in knowledge-intensive tasks, such as visual entity grounding (Hu et al., 2023) and information-seeking visual question answering (Chen et al., 2023d), where VLMs must ef-

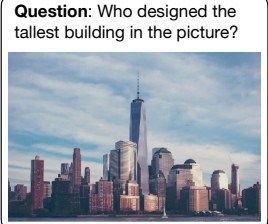

**Question**: Who designed the tallest building in the picture?

**Answer**: David Child

**Background Knowledge**

**Entity**: Freedom Tower

**Retrieved Content**: Freedom Tower is the main building of the rebuilt World Trade Center complex in Lower Manhattan, designed by David Childs of SOM.

Figure 1: An example question for information-seeking visual question answering.

fectively link the visual objects and scenes to their corresponding entities and relevant background knowledge. For instance, as illustrated in Figure 1, given the question "*Who designed the tallest building in the picture?*" alongside an image of several buildings, VLMs need to accurately iden-

tify the building based on its visual attributes and retrieve the associated background knowledge encoded in LLMs. However, the vast and dynamic nature of visual knowledge in the open world makes it impractical for VLMs to store all possible associations between visual appearances and their corresponding entities and background knowledge in their parameters.

One promising solution is retrieval-augmented generation (RAG), which integrates knowledge retrieved from external sources with VLMs and has demonstrated success in improving text-based knowledge-intensive tasks for LLMs (Guu et al., 2020; Lewis et al., 2020; Yoran et al., 2023). However, extending RAG to vision-language tasks presents several unique challenges: (1) **Modality Discrepancy**: Vision-language tasks usually rely on both visual and textual information, and neither modality can fully substitute for the other due to their inherent discrepancy, thus formulating precise retrieval queries is often difficult. For instance, textual inputs, such as "*Who designed the tallest building in the picture?*", usually contain generic terms or anaphoric references ("*the tallest building*") that lack specificity without visual context while visual information alone may not sufficiently clarify the query's intent, leading to ambiguity. This interplay between modalities makes it challenging to precisely retrieve relevant information from external sources. (2) **Information Noise**: Retrieved multimodal knowledge snippets, particularly those containing both images and text, often introduce irrelevant or extraneous information. A common type of noise arises when the primary entity in the retrieved image differs from the entity in the query image, leading to the retrieval of irrelevant textual knowledge. Another source of noise occurs within the retrieved images themselves, where background elements or unrelated objects, such as *Brookfield Place* in Figure 1, may distract the VLMs during perception and reasoning. This extraneous information can mislead the model and reduce the accuracy of its response to the query.

To tackle these challenges, we introduce RORA-VLM, a robust retrieval-augmented framework aiming at enhancing vision-language models on knowledge-seeking tasks. RORA-VLM consists of three novel components, each of which is tailored to address a unique challenge outlined above. To mitigate the modality discrepancy, we design IMAGE-ANCHORED TEXTUAL-QUERY EXPANSION, a 2-stage retrieval method that synergistically integrate vision-language information for more accurate and comprehensive vision-language retrieval. In the first stage, the query image, serving as a visual anchor, is used to retrieve visually similar images. For each retrieved image, we extract its associated entity name and brief description to augment the textual query and disambiguate the anaphoric references. The expanded query is then employed in the second stage to accurately retrieve the most relevant answers from a textual knowledge base. This 2-stage retrieval process ensures that the retrieved content is comprehensive and closely aligned with the multimodal query, minimizing the risk of incomplete or modality-restricted results. With this 2-stage retrieval process, we obtain multiple multimodal knowledge snippets to augment the VLMs, where each multimodal knowledge snippet is the concatenation of an image and entity description from the first stage and its corresponding retrieved texts from the second stage.

To further address the challenges posed by irrelevant information in retrieved multimodal knowledge snippets, we propose a two-fold approach, NOISE-RESILIENT RETRIEVAL-AUGMENTED GENERATION. First, we introduce an adversarial noise injection training strategy for robust augmentation, which encourages VLMs to selectively utilize retrieved knowledge for generation. Specifically, we construct training instances by intentionally introducing irrelevant information into the retrieved knowledge, compelling the model to become resilient to noises. By fine-tuning VLMs on a small number of instances of knowledge-intensive tasks, the model implicitly learns to compare visual nuances between the query image and retrieved images, thereby discarding irrelevant knowledge associated with images containing non-matching entities. Second, to handle the extraneous visual information, such as background objects or unrelated entities in images, we design a query-oriented visual token refinement strategy. VLMs typically encode each input image into a sequence of $n$ visual tokens via a CLIP image encoder (Radford et al., 2021) and each token corresponds to a distinct image patch. We refine the visual tokens of the query image by only keeping $M$ tokens ($m \ll n$) that are most related to the text query based on their CLIP embeddings, and similarly, for each retrieved image, we also identify and only keep the most relevant $m$ tokens to the query image.

We conduct extensive experiments to evaluate the effectiveness and robustness of our proposed framework on three widely adopted knowledge-seeking benchmarks: OVEN (Hu et al., 2023), InforSeek (Chen et al., 2023d), and Enc-VQA (Mensink et al., 2023). Our results demonstrate that, with only a minimal number of training instances (e.g., 10,000), the framework achieves significant improvements over baseline models, yielding up to 14.36% accuracy improvement, and consistently

outperforms Wiki-LLaVA (Caffagni et al., 2024), the current state-of-the-art retrieval-augmented VLM. Additionally, our extensive analysis reveals that: (1) The IMAGE-ANCHORED TEXTUAL-QUERY EXPANSION method comprehensively leverages multimodal information to enhance query intent understanding and improve retrieval accuracy in knowledge-intensive tasks, achieving up to an 11.52% increase in retrieval precision compared to the general single-stage retrieval approach. (2) The NOISE-RESILIENT RETRIEVAL-AUGMENTED GENERATION enables VLMs to identify valuable information relevant to the entity in query image from the retrieval knowledge and focus on visual tokens that are closely related to the entity concerned in the input text query. (3) Pre-training on entity-rich image-caption pairs (Burns et al., 2023) substantially enhances the VLMs' performance on information-seeking VQA tasks. (4) RORA-VLM also demonstrates strong zero-shot transfer to knowledge-intensive tasks from unseen domains.

## 2 RELATED WORK

**Vision-Language Models** Recent advancements in vision-language models (VLMs), such as BLIP-2 (Li et al., 2023), Flamingo (Alayrac et al., 2022), LLaVA (Liu et al., 2023b), and Instruct-BLIP (Dai et al., 2023), have demonstrated remarkable performance on various visual perception tasks, such as image captioning (Lin et al., 2014; Schuhmann et al., 2022; Chen et al., 2023a), visual question answering (Antol et al., 2015; Marino et al., 2019; Schwenk et al., 2022), object detection (Lin et al., 2014; Everingham et al.), visual grounding (Hu et al., 2023; Kazemzadeh et al., 2014), and visual relationship detection (Lu et al., 2016), etc. These models typically employ an architecture consisting of a pre-trained visual encoder (Radford et al., 2021; Dosovitskiy et al., 2021; Chen et al., 2024), a pre-trained large language model (Touvron et al., 2023; Almazrouei et al., 2023), and a projection function that maps visual features to the text embedding space (Liu et al., 2023b). However, this method often falls short in aligning visual features with the extensive knowledge embedded in language models. Alternative architectures, such as the Q-former used in BLIP-2 (Li et al., 2023) and the perceiver resampler in Flamingo (Alayrac et al., 2022), have been proposed to enhance the perception of visual content. These architectures focus on improving the models' ability to understand the color, shape, and layout of objects and scenes. Despite these advancements, VLMs still struggle with knowledge-intensive tasks that require deep integration of visual and textual information. This gap highlights the need for more sophisticated methods to align visual features with the rich semantic knowledge stored in language models.

**Retrieval-Augmented Generation Work** Augmenting models with external knowledge sources has proven effective in enhancing their performance on knowledge-intensive tasks. In the text-only domain, models like REALM (Guu et al., 2020), RAG (Lewis et al., 2020), and RobustRAG (Yoran et al., 2023) have demonstrated the benefits of retrieval-based augmentation. These models retrieve relevant information from external sources to provide additional context for generating accurate responses. Applying retrieval-augmented generation to the vision-language domain presents unique challenges due to modality discrepancies and differing model architectures (Wei et al., 2023). Several recent studies (Gui et al., 2021; Lin et al., 2023; 2024) have explored multimodal retrieval to enhance LLMs by retrieving textual knowledge from visual queries. However, they primarily focus on improving retrieval quality, while our research focuses more on addressing the fundamental challenge of how to effectively and robustly leverage external knowledge to augment the reasoning and generation of vision-language models. Given that the state-of-the-art retriever can only achieve modest performance, e.g., lower than 0.2 for recall@1 on InfoSeek (Chen et al., 2023d), managing and denoising the noise becomes more crucial for VLMs. This work distinctively addresses this challenge by introducing a robust retrieval augmentation framework. Our proposed RORA-VLM framework distinctively addresses this challenge by mitigating retrieval-induced noise while enhancing VLMs' ability to handle interleaved visual-textual contexts, ultimately improving generalizability to unseen entities, events, and scenes.

**Knowledge-Intensive Tasks and Benchmarks** Knowledge-intensive tasks pose significant challenges for VLMs, requiring them to connect visual appearances with semantic knowledge and perform complex reasoning. Benchmarks such as OVEN (Hu et al., 2023) and InfoSeek (Chen et al., 2023d) have been developed to evaluate VLMs on tasks like visual entity grounding and information-seeking visual question answering. For instance, tasks like identifying the designer of a building from an image require VLMs to recognize the building based on its visual properties and infer the designer using stored knowledge. Studies have shown that extensive fine-tuning on knowledge-intensive task instances does not substantially improve VLMs' performance (Chen

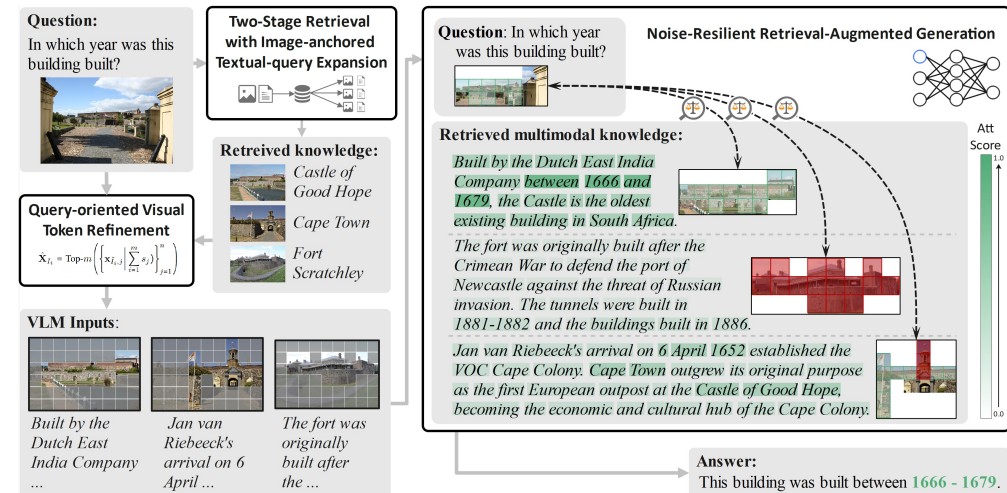

Figure 2: Overview of architecture of RORA-VLM.

et al., 2023d; Hu et al., 2023; Mensink et al., 2023). This indicates that current architectures are not sufficiently equipped to handle the dynamic and detailed nature of visual-semantic associations. Our RORA-VLM framework aims to bridge this gap by explicitly aligning visual features with internal knowledge and augmenting VLMs with external knowledge sources. By integrating both visual and textual information more effectively, RORA-VLM seeks to improve VLMs' capabilities on knowledge-intensive tasks and set new benchmarks for performance in this domain.

## 3 METHOD

### 3.1 PROBLEM FORMULATION

In this work, we mainly focus on improving VLMs on knowledge-intensive VQA tasks via retrieval-augmented generation. Given a text query $q$ together with an image $I$, a VLM is expected to generate a response $y$ by leveraging the multimodal knowledge snippets $\mathbf{R}$ retrieved from an external database as context. The objective of the retrieval-augmented generation can be formulated as:

$$y = \arg\max_y P(y|q, I, \mathbf{R}), \tag{1}$$

Figure 2 depicts an overview of our proposed framework, RORA-VLM, which consists of two novel designs to robustly enhance VLMs with retrieval augmentation: (1) two-stage vision-language retrieval with Image-anchored Textual-query Expansion; and (2) Noise-Resilient Retrieval-Augmented Generation. We detail each design as follows.

### 3.2 IMAGE-ANCHORED TEXTUAL-QUERY EXPANSION

Queries in knowledge-intensive VQA tasks typically consist of complementary visual and textual information—query images highlight the key entities concerned in the question, while query texts express the intent of the question using generic terms or anaphoric references to those entities. To comprehensively leverage the combined visual and textual information in the queries and retrieve relevant knowledge effectively, we design a 2-stage retrieval process with image-anchored textual-query expansion as follows. Figure 6 in Appendix A.10 also provides a detailed illustration of the 2-stage retrieved process.

**Stage-1: Image-anchored Entity Retrieval** In this stage, we utilize the input query image $I$, as an anchor, to retrieve visually similar images $\tilde{\mathbf{I}}_{re} = \{\tilde{I}_1, \tilde{I}_2, ...\}$ from an image database. Specifically, the image database is built upon WIT (Srinivasan et al., 2021) which contains 37.6 million entity-rich image-text pairs, with each text providing the name and background information of the entity depicted in the image, sourced from Wikipedia. To enable efficient retrieval, we encode each image in WIT into a vector using the CLIP (Radford et al., 2021) image encoder, and construct a dense

vector-search database[1]. In this database, the encoded image features $\mathbf{z}_i = \text{CLIP}(\tilde{I}_i) \in \mathbb{R}^d$, where $d$ is the dimension of the CLIP embedding, serve as search indexes $\mathbf{Z} = \{\mathbf{z}_1, \mathbf{z}_2, \ldots, \mathbf{z}_N\}$, while the corresponding entity names and background information for these images are stored as search values $\mathbf{E} = \{e_1, e_2, \ldots, e_N\}$, where $e_i$ denotes the entity name and background information for candidate image $\tilde{I}_i$ and $N$ is the total number of entries in the database. Given a query image $I$, the image retriever $\phi^{\text{img}}$ computes the cosine similarity between the query image and all search indexes using their CLIP embeddings and fetches the top-$k$ most similar images along with their associated entity name and background information. More details of the image retriever are provided in Appendix A.1.

**Stage-2: Query-expanded Text Retrieval**    With the entity name and description of each retrieved image from the first stage, we further use them to expand the original text query and develop the second stage query-expanded text retrieval with a *Google Search*[2] engine, leveraging the vast resources of the web to enhance retrieval accuracy. Specifically, given the original text query $q$ and a retrieved entity name and description $e_i$, the text retriever $\phi^{\text{txt}}$ searches for top-$l$ textual knowledge snippets that are most relevant to the expanded query:

$$\mathbf{c}_i = \{c_{i,1}, c_{i,2}, \ldots, c_{i,l}\} = \phi^{\text{txt}}(q, e_i), \tag{2}$$

where $\mathbf{c}_i$ denotes the set of textual knowledge snippets related to the entity description $e_i$ and the retrieved image $\tilde{I}_i$.

Finally, we concatenate each retrieved image $\tilde{I}_i$ from the first stage and the corresponding textual knowledge snippets $\mathbf{c}_i$ from the second stage as a sequence $\mathbf{r}_i = [\tilde{I}_i : \mathbf{c}_i]$, where : denotes the concatenation operation, and obtain the multimodal knowledge snippets $\mathbf{R} = \{\mathbf{r}_1, \mathbf{r}_2, \ldots, \mathbf{r}_k\}$ to later augment the VLMs.

### 3.3 Noise-Resilient Retrieval-Augmented Generation

Since the retrieving process is not perfect, the retrieved multimodal knowledge snippets may contain irrelevant information to the given query. In this section, we present Noise-Resilient Retrieval-Augmented Generation, a two-fold denoising approach that enables VLMs to selectively utilize the retrieved knowledge for answer prediction and ignore irrelevant retrieval noise.

**Adversarial Noise Injection for Robust Augmentation**    For training, we design a adversarial noise injection for robust augmentation that intentionally introduces irrelevant information into the retrieved knowledge, forcing the model to be robust to noises when leveraging the retrieved knowledge for answer prediction. For each training instance, i.e., a text query alongside an image $(I, q)$, of the knowledge-intensive VQA task, we first retrieve the top-$(k\text{-}1)$ multimodal knowledge snippets $\mathbf{R} = \{\mathbf{r}_1, \mathbf{r}_2\}$ and randomly sample an irrelevant knowledge snippet[3] $\mathbf{r}' = [\tilde{I}' : \mathbf{c}']$ from the retrieval database. We then concatenate them together with the original query to form a sequence of interleaved images and text: $[\mathbf{r}_1 : \mathbf{r}_2 : \mathbf{r}' : I : q]$, which is further fed as input to VLMs for answer prediction. We fine-tune VLMs on such retrieval-augmented training instances with noise and minimize the cross-entropy loss of predicting the target answers.

**Query-oriented Visual Token Refinement**    Images often contain much noise, such as objects or visual scenes that are unrelated to the concerned entities. To further filter out query-irrelevant visual information within the retrieved images as well as the query image, we design a *query-oriented visual token refinement* strategy[4]. Given a text query $q$ alongside a query image $I$, we first encode the text query using the CLIP text encoder, producing a text embedding $\mathbf{x}_q \in \mathbb{R}^d$, where $d$ is the embedding dimension. Similarly, the image $I$ is encoded into a sequence of visual embeddings $\mathbf{X}_I = \{\mathbf{x}_{I,1}, \mathbf{x}_{I,2}, \ldots, \mathbf{x}_{I,n}\} \in \mathbb{R}^{n \times d}$, where $\mathbf{x}_{I,i} \in \mathbb{R}^d$ denotes a visual token embedding corresponding to

---

[1]We construct the vector-search database based on a hierarchical navigable small-world (HNSW) graph (Malkov & Yashunin, 2018).

[2]We query Google search via the Serper service: https://serper.dev/

[3]We randomly sample an entity from our retrieval database, together with its image and corresponding knowledge, as the irrelevant sample. We make sure the sampled entity is mismatched with the target entity.

[4]Figure 7 in Appendix A.11 provides an example to illustrate the query-oriented visual token refinement process.

an image patch, and $n$ is the number of visual tokens. The details of the encoding process can be found in Appendix A.2. For each visual token embedding $\mathbf{x}_{I,i}$, we calculate its similarity to the text embedding by dot product: $s_i = \mathbf{x}_{I,i} \cdot \mathbf{x}_q$. Then, the top-$m$ most similar visual tokens are selected, forming the refined visual token sequence $\hat{\mathbf{X}}_I \in \mathbb{R}^{m \times d}$ of the query image:

$$\hat{\mathbf{X}}_I = \text{Top-}m\left(\left\{\mathbf{x}_{I,i} \middle| s_i\right\}_{i=1}^n\right). \tag{3}$$

Similarly, we also encode each of the retrieved image $\tilde{I}_i \in \tilde{\mathbf{I}}_{\text{re}}$ into a sequence of visual token embeddings $\mathbf{X}_{\tilde{I}_i} = \{\mathbf{x}_{\tilde{I}_i,1}, \mathbf{x}_{\tilde{I}_i,2}, ..., \mathbf{x}_{\tilde{I}_i,n}\} \in \mathbb{R}^{n \times d}$. For each visual token embedding $\mathbf{x}_{\tilde{I}_i,j} \in \mathbb{R}^d$, we compute its similarity to the query image by calculating the sum of its dot product with all of the selected visual tokens of the query image: $s_j = \sum_{i=1}^m (\mathbf{x}_{I,i} \cdot \mathbf{x}_{\tilde{I}_i,j})$ where $\mathbf{x}_{I,i} \in \hat{\mathbf{X}}_I$. Then, the top-$m$ most relevant visual tokens of the retrieved image are selected, forming the refined visual token sequence $\hat{\mathbf{X}}_{\tilde{I}_i} \in \mathbb{R}^{m \times d}$ for each of the query image:

$$\hat{\mathbf{X}}_{I_i} = \text{Top-}m\left(\left\{\mathbf{x}_{\tilde{I}_i,j} \middle| \sum_{i=1}^m s_j\right\}_{j=1}^n\right). \tag{4}$$

## 4 EXPERIMENT SETUP

**Evaluation Benchmarks** To evaluate the effectiveness and robustness of RORA-VLM, we conduct experiments on three benchmark datasets, including OVEN (Hu et al., 2023) for visual entity grounding, and InfoSeek (Chen et al., 2023d) and Encyclopedic-VQA (Mensink et al., 2023) for information-seeking visual question answering. As the test sets of OVEN and InfoSeek are not available at the time of submission, we report our results on their validation sets. More details of these datasets can be found in Appendix A.8.

**Evaluation Metrics** We adopt evaluation metrics in line with previous studies (Hu et al., 2023; Chen et al., 2023d; Mensink et al., 2023). For visual entity recognition task, we use the standard *accuracy* metric to assess the model's capability to correctly identify entities in images. For knowledge-seeking visual question answering (VQA) task, we apply two different metrics tailored to the specific types of questions. For questions expecting a string-based response, such as entity names, we report accuracy using the *VQA accuracy* metric (Antol et al., 2015). This metric allows for multiple valid answers by considering slight variations in phrasing (e.g., "New York City" and "NYC") as correct. The model is evaluated based on whether its answer matches any of these valid responses. For questions requiring numeric answers, we use *relaxed accuracy* (Methani et al., 2020), which accounts for small deviations from the exact numerical value. This metric considers an answer correct if it falls within an acceptable tolerance range around the ground truth.

**Baselines** We compare our framework with several state-of-the-art vision-language models. **LLaVA-v1.5** (Liu et al., 2023a) integrates pre-trained visual and language models for strong performance in multimodal tasks, while **LLaVA-v1.6** (Liu et al., 2024) introduces improved fine-tuning techniques. **PaLI-17B** (Chen et al., 2023c) utilizes a 17-billion-parameter architecture, excelling in image captioning and visual question answering, with **PaLI-X** (Chen et al., 2023b) improving performance on vision-language tasks by scaling up the model size and incorporating a high-capacity visual encoder. **BLIP-2** (Li et al., 2023) introduces efficient visual grounding through a Q-former, and **InstructBLIP** (Dai et al., 2023) enhances it for instruction-following tasks. **CLIP2CLIP** (Hu et al., 2023) leverages a CLIP-based model for improved image captioning. Recent work **Wiki-LLaVA** (Caffagni et al., 2024) is designed for entity-centric question answering, aligning visual data with external knowledge from Wikipedia. **PreFLMR** Lin et al. (2024) introduces a robust multimodal retriever pre-trained on a vision-language corpus comprising over ten million samples, enabling high-quality retrieval to augment the generation processes. **RA-CM3** Yasunaga et al. (2023) employs a cross-modality retrieval mechanism to access and leverage multimodal information to enhance the performance of multimodal generation. To ensure a fair comparison, all the baseline models are fine-tuned on the OVEN Hu et al. (2023), InfoSeek Chen et al. (2023d), and Enc-VQA Mensink et al. (2023) datasets respectively, and then evaluated on the corresponding tasks.

**Model Tuning** Building on the pre-trained VLMs, we conduct an additional **visual-knowledge alignment pre-training** on a knowledge-intensive multimodal dataset WikiWeb2M (Burns et al.,

2023). We curated 1 million entity-rich image-text instances from the WikiWeb2M, and each instance consists of a unique image depicting an entity, its corresponding image caption, and the title and main content of the section associated with that image. For the training process, we treat each image-text instance as a single-turn conversation by randomly sampling a language instruction $\mathbf{X}_q$ from a pre-defined instruction pool, prompting the model to caption the image and provide background knowledge. The input for each training instance consists of an image and a query. The ground-truth answer is formed by concatenating the original caption, section title, and section content. To align the visual appearance of entities and their background knowledge stored in LLM, we only freeze the weights of the visual encoder during the training and optimize the parameters of both the projection layer and the LLM. After pre-training, for each of the OVEN (Hu et al., 2023), InfoSeek (Chen et al., 2023d), and Encyclopedic-VQA (Mensink et al., 2023) datasets, we further randomly sampled 1,000 instances to perform a **lightweight fine-tuning** of the VLM on these subsets for specific downstream knowledge-intensive VQA tasks. More implementation details are shown in Appendix A.9

## 5 RESULT & DISCUSSION

**Main Results** Table 1 presents the main results for visual entity grounding on the OVEN dataset and information-seeking visual question answering on the InfoSeek and Encyclopedic-VQA datasets. Though only with 7B parameters and fine-tuned on less than 10,000 instances per dataset, RORA-VLM significantly outperforms all baselines that are with much larger model sizes (including 17B and 55B models) and fine-tuned on substantially more instances (i.e., up to 1 million) across nearly all benchmarks, except for the Query subset of the OVEN dataset. The Query subset

Table 1: Evaluation results in accuracy (%). The best performance is highlighted in **bold**.The Entity groups expect an entity name as the target answer, while Query groups target a general object name or concept as the answer. * denotes our implementation of Wiki-LLaVA as its original source code is not publicly available.

| Model | Size (B) | OVEN | | InfoSeek | | Enc-VQA |
|---|---|---|---|---|---|---|
| | | Entity | Query | Entity | Query | |
| CLIP2CLIP | 0.86 | 10.10 | 2.10 | - | - | - |
| PaLI | 17 | 12.40 | 22.40 | 16.00 | 20.70 | - |
| PaLI-X | 55 | 20.80 | 23.50 | - | - | - |
| BLIP-2 | 12 | - | - | 13.30 | 14.50 | - |
| InstructBLIP | 12 | - | - | 13.20 | 14.30 | - |
| RA-CM3 | 7 | - | - | 17.09 | 21.64 | - |
| PreFLMR | 7 | - | - | 19.37 | 22.21 | - |
| LLaVA-v1.6 | 7 | 3.72 | **24.55** | 14.16 | 15.98 | 13.54 |
| LLaVA-v1.5 | 7 | 3.63 | 20.04 | 10.34 | 12.98 | 12.21 |
| Wiki-LLaVA* | 7 | 14.43 | 20.4 | 21.44 | 23.68 | 18.61 |
| RORA-VLM | 7 | **15.08** | 24.06 | **25.10** | **27.34** | **20.29** |

of OVEN primarily focuses on visual perception questions (e.g., "What is in the bowl?" with the answer "egg") that require less reliance on fine-grained entity knowledge (Hu et al., 2023). Compared to our base model LLaVA-v1.5, LLaVA-v1.6 enhances its capacity to better perceive details in images with higher-resolution image inputs and is trained on large-scale visual perception datasets (Chen et al., 2023a). In contrast, our approach focuses on robust retrieval augmentation for tasks that depend heavily on entity background knowledge, thus improving the visual perception capabilities of VLMs is beyond the scope of our work.

**Effect of Query-oriented Visual Token Refinement** We conduct an ablation study to demonstrate the effectiveness of Query-oriented Visual Token Refinement, with the results presented in Table 2. In the "w/o VK-Refinement" setting, we use the widely adopted average pooling (kernel size of 2, stride of 2) to obtain the same number of visual tokens as our refinement approach. The details of the pooling process can be found in Appendix A.3. As we can see, without explicitly filtering out the irrelevant visual information, the performance drops on both subsets of InfoSeek. In Figure 3, we show the qualitative results of the Query-oriented Visual Token Refinement method. From the query image, we select $m$=144

Table 2: Ablation studies for query-oriented visual token refinement (w/o VK-Refinement) and noise-resilient retrieval-augmented generation (text-only RAG) on InfoSeek. Performance is reported in accuracy (%).

| Model | Entity | Query |
|---|---|---|
| RORA-VLM(ours) | 24.56 | 26.33 |
| - w/o VK-Refinement | 23.94 | 24.85 |
| - text-only RAG | 17.29 | 19.28 |

visual tokens that are most related to the text query (i.e., the Question), while each visual token corresponds to an image patch (highlighted in yellow). As we can see, this method effectively identifies and selects patches corresponding to the key visual entity, even with the presence of anaphoric

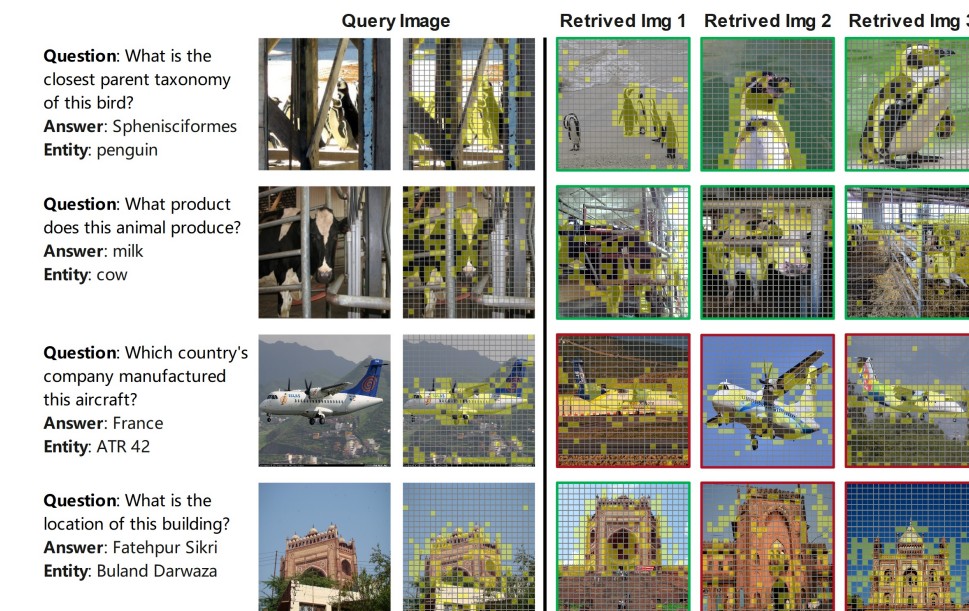

Figure 3: Qualitative results for query-oriented visual token refinement.

references in the query. Similarly, for each retrieved image, we also select $m=144$ visual tokens that are most related to the query image. For retrieved images containing the same entities as the query image (highlighted with a green box), the selected patches tend to cluster around the key entity. Conversely, when retrieved images contain different entities from those in the query image (highlighted with a red box), the distribution of selected patches is more scattered. These qualitative results underscore the effectiveness of our token refinement strategy in filtering out irrelevant visual information, enabling the retrieval augmentation of VLMs more robust.

**Effect of Adversarial Noise Injection for Robust Augmentation**   The essential assumption of adversarial noise injection for robust augmentation is that by training with adversarial noise, VLMs implicitly learn to compare the visual appearances of entities in the retrieved images and the query image, thereby discarding irrelevant information from the textual knowledge snippets corresponding to the irrelevant retrieved images. To validate this assumption and demonstrate that the performance improvement is not solely due to the retrieved textual knowledge, we remove the retrieved images from the multimodal knowledge snippets in RoRA-VLM during both training and inference, while keeping all other hyperparameters and the 2-stage retrieval process identical. In this configuration, the multimodal knowledge snippets are reduced to textual-only knowledge snippets, and we refer to this setting as RoRA-VLM with textual-only RAG. During training, the model can still leverage the retrieved textual knowledge to answer questions; however, without the presence of images, it cannot learn to differentiate the relevance of the textual knowledge based on the visual appearances of entities. As shown in Table 2, without retrieved images to provide visual cues for selecting relevant knowledge, RORA-VLM with textual-only RAG exhibits significantly worse performance compared to the standard RORA-VLM, despite having access to the same textual knowledge during inference. Additionally, we analyze the robustness of RORA-VLM under varying levels of retrieval noise, with the results presented in Appendix A.4.

To complement our findings on retrieval noise and better understand how RORA-VLM prioritizes relevant information during inference, we visualize the attention scores assigned to each input token during answer generation. As shown in Figure 4, the left column presents the input queries, images, target answers, and RORA-VLM's predictions. The middle column displays the retrieved images along with their associated textual knowledge. The green highlights indicate the model's attention to individual tokens, with darker shades denoting higher attention scores. The right column provides a detailed breakdown of the attention distribution, with gray bars representing the positions of the retrieved images. By examining these qualitative results, we observe that RORA-VLM effectively learns to focus on the textual knowledge corresponding to images containing entities that match those in the query image. For instance, in the second row of Figure 4, RORA-VLM predom-

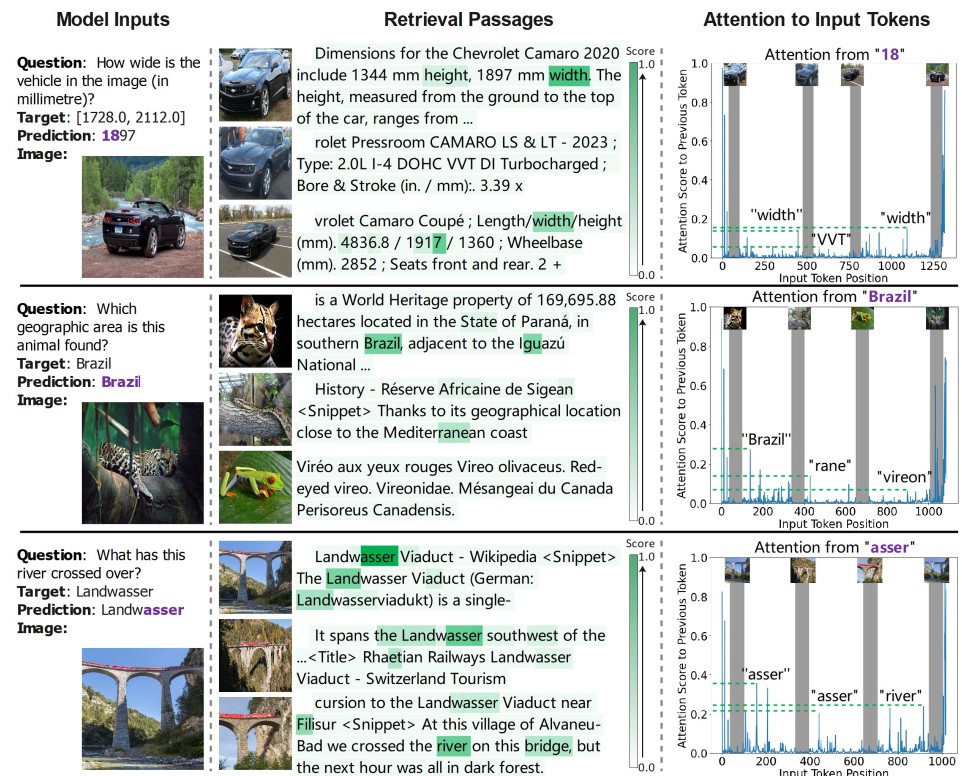

Figure 4: Visualization of attention scores assigned to VLM input tokens during next-token generation. Tokens are highlighted in green, with darker shades indicating higher attention scores.

inantly focuses on the first two knowledge snippets, while disregarding the third, which pertains to a completely different animal.

**Effect of Knowledge-Intensive Pre-training** To demonstrate the effectiveness of our proposed knowledge-intensive pre-training, we design two sets of experiments and report the results in Table 3. For the original LLaVA-v1.5 without retrieval augmentation, by performing knowledge-intensive pre-training, the performance is significantly improved on both InfoSeek. Similar improvements are also observed by comparing RORA-VLM to RORA-VLM w/o WikiWeb2M. Additionally, we compare pre-training dataset between WikiWeb2M and ShareGPT4V (Chen et al., 2023a)), a

Table 3: Performance in accuracy (%) for VLMs with or without knowledge-intensive pre-training on InfoSeek.

| Model | Entity | Query |
|---|---|---|
| LLaVA-v1.5 | 10.34 | 12.98 |
| - w/ WikiWeb2M | 18.00 | 20.98 |
| RORA-VLM (ours) | 24.56 | 26.33 |
| - w/o WikiWeb2M | 20.68 | 23.41 |
| - w/ ShareGPT4V | 21.28 | 22.84 |

generic image-caption dataset where the captions only describe the image context without many fine-grained entities or entity descriptions. As we shown, the performance of RORA-VLM w/ ShareGPT4V is much lower than RORA-VLM pre-trained on WikiWeb2M, demonstrating the benefit of knowledge-intensive pre-training on better aligning the visual appearance of objects to their corresponding entities and entity background knowledge.

**Domain Transfer Capability** In this subsection, we examine the generalizability of the proposed RORA-VLM using the Encyclopedic-VQA dataset. The iNaturalist subset of the Encyclopedic-VQA dataset consists of questions concerning 11 categories (e.g., Plant, Insect, Lake, etc.) of entities. To create a domain transfer setting, we select "Insect"

Table 4: Performance in accuracy (%) for domain transfer on Encyclopedic-VQA.

| Model | SFT | Domain Transfer |
|---|---|---|
| LLaVA-v1.5 | 18.23 | 17.18 |
| RORA-VLM(ours) | 24.36 | 20.26 |

as the target domain, and modify the training set by filtering out instances from the "Insect" category. We fine-tune both the baseline model and our RORA-VLM on the original training set of the iNaturalist subset as well as the modified training set for domain transfer, and evaluate on the

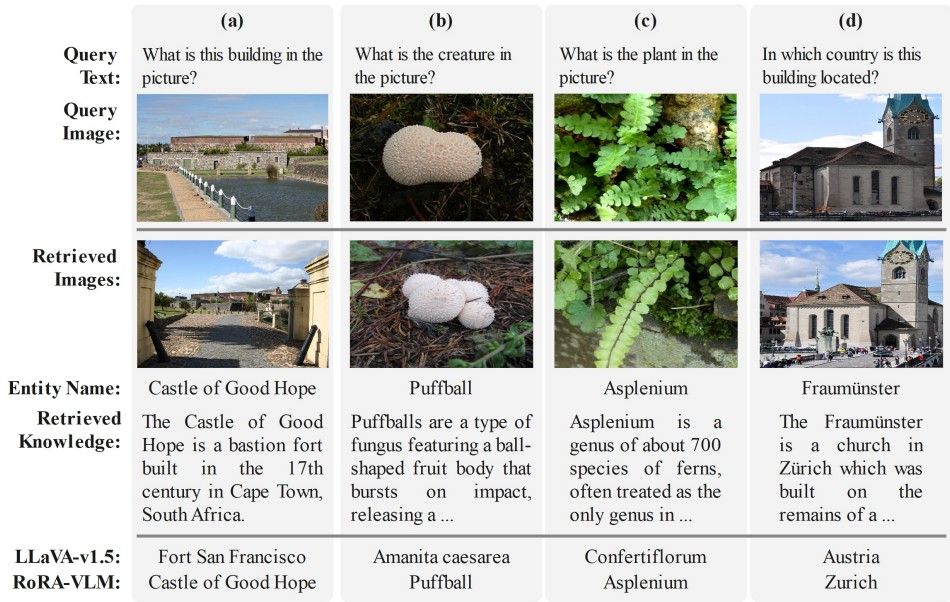

Figure 5: Qualitative results of 2-stage retrieval with image-anchored textual-query expansion.

complete test set of the iNaturalist subset. Table 4 shows the results, where "SFT" refers to models fine-tuned on the full training set, while "Domain Transfer" refers to models fine-tuned on the modified training set for domain transfer. The results clearly show that, even without being fine-tuned on the "Insect" category, RORA-VLM still outperforms the baseline model that is trained on the complete training set. This demonstrates the generalizability of our proposed method, as it enables the VLM to surpass its base model even without access to in-domain knowledge during training.

**Evaluation of the Two-Stage Retrieval** We report the retrieval precision at each stage of our proposed two-stage retrieval process in Table 5. In the first stage, given a query image, if the target entity shown in the query image matches any of the retrieved $m$ images, we take it as correct. Similarly, in the second stage, if the golden answer is included in any of the re-

Table 5: Retrieval precision (%) for the first and second stage of retrieval.

| Stage | OVEN | | InfoSeek | |
|---|---|---|---|---|
| | Entity | Query | Entity | Query |
| First Stage | 35.16 | 34.45 | 38.53 | 37.67 |
| Second Stage | - | - | 27.01 | 26.97 |

trieved textual knowledge snippets, we also view it as correct. Figure 5 presents several examples for qualitative analysis. Our retrieval method effectively identifies images that contain entities matching those in the query images. Although the perspectives of the entities in the retrieved images differ from those in the query images, the retrieved images provide sufficient visual attributes for entity identification (e.g., the gap in the wall in Figure 5(a) and the shape of the leaves in Figure 5(c)). Additionally, we performed an ablation experiment using only a single-stage retrieval method to emphasize the effectiveness of our two-stage retrieval approach, with the results presented in Appendix A.5.

## 6 CONCLUSION

In this work, we introduce RORA-VLM, a novel and robust retrieval-augmented framework specifically designed for VLMs to address two key challenges: (1) the intrinsic discrepancy between multimodal queries, and (2) the presence of irrelevant and extraneous information embedded in the retrieved multimodal knowledge snippets. RORA-VLM incorporates two technical innovations: (1) a two-stage retrieval process with image-anchored textual-query expansion that synergistically integrates visual and textual information for more comprehensive retrieval results, and (2) a robust retrieval augmentation method that enhances the VLMs' resilience against noise. Our experimental results demonstrate that RORA-VLM achieves state-of-the-art performance on three widely adopted benchmark datasets, including OVEN, InfoSeek, and Enc-VQA.

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

# A APPENDIX

## A.1 IMAGE-ANCHORED ENTITY RETRIEVAL

In this stage, we utilize the input query image $I$, as an anchor, to retrieve visually similar images $\tilde{\mathbf{I}}_{re} = \{\tilde{I}_1, \tilde{I}_2, ...\}$ from an image database. The image retriever $\phi^{img}$ leverages a non-parametric function to measure the cosine similarity between the CLIP embedding of query image $I$ and all search indexes. The score of each candidate image $\tilde{I}_i$ with search index $\mathbf{z}_i$ can be expressed as:

$$P(\tilde{I}_i|I, \mathbf{Z}) = \frac{\exp\left(\text{Sim}(I, \mathbf{z}_i)\right)}{\sum_{j=1}^{n} \exp\left(\text{Sim}(I, \mathbf{z}_j)\right)}, \text{Sim}(I, \mathbf{z}_i) = \frac{\text{CLIP}(I)^{\top}\mathbf{z}_i}{\|\text{CLIP}(I)\| \|\mathbf{z}_i\|} \tag{5}$$

Based on this function, the image retriever $\phi^{img}$ fetches the top-$k$ images that are most similar to the query image along with their associated entity name and background information.

$$\{(\tilde{I}_1, e_1), (\tilde{I}_2, e_2), \ldots, (\tilde{I}_k, e_k)\} = \phi^{img}(I, \mathbf{Z}, \mathbf{E}), \tag{6}$$

## A.2 DETAILS OF THE CLIP MODEL ENCODING

In this section, we provide a detailed description of how we encode an image into a sequence of visual embeddings using CLIP.

**Image Encoding with CLIP:** In the CLIP model, the visual encoder is based on the Vision Transformer (ViT) architecture. Given an image, the visual encoder processes it as a whole and encodes it into a feature representation of shape [576, 1024]. This representation can be interpreted as 576

vectors, each with a dimensionality of 1024. The 576 vectors correspond to patches of the input image, where the image is internally divided into a grid of patches during the encoding process. This division is not explicit; rather, it is an inherent part of the ViT architecture, which computes patch-level embeddings directly through a convolutional embedding layer applied to the full image. The resulting intermediate patch embeddings collectively form the image's representation in the model's latent space.

**Dimensionality of Visual Embeddings:** After passing through the vision transformer (ViT) layers, each patch is represented as a feature vector with a dimensionality of 1024. To further process these features, we utilized the final visual projection layer of the original CLIP model. This projection layer, which is also used for the pooled [CLS] token in the original implementation, is applied to all 576 patch-based feature vectors in our approach. The projection reduces the dimensionality of each feature vector from 1024 to 768. To clarify further, the visual projection layer is part of CLIP's original implementation. While it is typically applied only to the pooled [CLS] token to produce the image-level feature representation, in our work, we extend its application to all 576 patch-level feature vectors. As a result, the output is a feature representation of shape [576, 768], where 576 corresponds to the number of patches and 768 is the dimensionality of the projected patch embeddings.

After computing the patch embeddings, for each text query, we derive a 768-dimensional vector from the [CLS] token of the CLIP text encoder. We then compute the similarities between the text embedding and the image patch embeddings to select the top-m relevant patches, which are subsequently projected into the LLM's latent space using the LLaVA projector.

### A.3 DETAILS OF THE POOLING PROCESS

As detailed in the Appendix A.2, each image is processed into a feature matrix with shape [576, 768] by the CLIP visual encoder and the LLaVA projector. Our proposed Visual Token Refinement method further selects the top 144 visual tokens that are most relevant to the query, constructing a feature matrix of shape [144, 768]. This selection process enables the VLM to focus more effectively on query-relevant image content while mitigating the influence of irrelevant noise, such as image backgrounds or query-irrelevant entities present in the image. To conduct an ablation study of the Visual Token Refinement method, we replace it with a simple average-pooling-based baseline, which also takes in the original [576, 768] visual patch vectors as input, downsample and convert them into [144, 768] vectors to ensure a fair comparison with our Visual Token Refinement method. Specifically, we first reshape the first dimension of the feature matrix (i.e., 576) into a 2D grid with dimensions $24 \times 24$, corresponding to the spatial arrangement of patches in the original image, then apply a 2D average pooling operation with a kernel size of $2 \times 2$ and a stride of 2. This pooling reduces the spatial resolution from $24 \times 24$ to $12 \times 12$, yielding 144 patch vectors in total while each patch vector has a dimensionality of 768. By reducing the number of feature vectors from 576 to 144, this process ensures compatibility with the limited sequence length of the LLM and aligns the number of input tokens for the average pooling baseline with that of our visual token refinement method. This alignment allows for a direct and fair comparison of the two approaches in the ablation study.

### A.4 ROBUSTNESS OF RORA-VLM UNDER VARYING LEVELS OF RETRIEVAL NOISE

To further analyze the ability of our RoRA-VLM to handle noisy retrieval and validate its robustness, we conducted additional ablation studies involving controlled retrieval noise scenarios. The key challenge in ideally proving the effectiveness of our model in ignoring retrieval noise is the lack of gold-standard labels for the retrieval process in the evaluation datasets. Specifically, we do not have precise relevancy labels between input queries and all candidate samples for retrieval, making it infeasible to construct an experiment with exactly one relevant sample and two randomly sampled irrelevant samples. Therefore, we designed an alternative experiment with varying levels of retrieval noise. During the inference stage, instead of using the top-3 retrieved entity images and their corresponding knowledge snippets, we tested a setting where we used the top-1 retrieved entity image and its knowledge snippet along with two randomly sampled irrelevant entity images

and their knowledge snippets. This random sampling process was repeated twice, resulting in two distinct sets of irrelevant entity images and knowledge snippets for the same input instance. Additionally, we tested another setting using only the top-1 retrieved entity image and its corresponding knowledge snippet for generation augmentation. Using these four configurations of retrieved entity images and knowledge snippets, we evaluated retrieval augmentation on the InfoSeek dataset. The results are summarized in the Table 6. From the results, we observe that the model's performance remains relatively stable regardless of which two noise samples were chosen, demonstrating to some extent the model's ability to identify useful information from the retrieved samples while ignoring irrelevant ones. However, due to the absence of ground-truth labels for the retrieval process, there is no guarantee that the top-1 retrieval output is always correct. Consequently, it is reasonable to observe a slight performance degradation when irrelevant entities are used to replace the top-2 and top-3 retrieved samples. Moreover, when comparing the variant using only the top-1 retrieval for augmentation with the variants including irrelevant retrieval noise, we note that the inclusion of irrelevant samples does not significantly degrade overall performance. These results highlight the robustness of our method to retrieval noise and its ability to leverage relevant knowledge snippets for improved inference.

Table 6: Performance in accuracy (%) for RoRA-VLM with varying levels of retrieval noise on InfoSeek.

| Model | Entity | Query |
|---|---|---|
| Top-1 Retrieval | 20.49 | 22.19 |
| Top-1 Retrieval + 2 Noises (1) | 19.61 | 21.97 |
| Top-1 Retrieval + 2 Noises (2) | 19.63 | 22.02 |
| Top-3 Retrieval | 25.10 | 27.34 |

Table 7: Performance in accuracy (%) for VLMs with or without knowledge-intensive pre-training on InfoSeek.

| Model | Entity | Query |
|---|---|---|
| LLaVA-v1.5 | 10.34 | 12.98 |
| RA-CM3 (single-stage) | 17.09 | 21.64 |
| RoRA-VLM (single-stage) | 21.9 | 23.87 |
| RoRA-VLM (2-stage) | 25.10 | 27.34 |

## A.5 ABLATION STUDY ON SINGLE-STAGE RETRIEVAL

We performed an ablation experiment using only a single-stage retrieval method to emphasize the effectiveness of our two-stage retrieval approach. Specifically. In the single-stage configuration, we utilized the CLIP embedding of the query image to retrieve the most similar entity images in our retrieval database, and thereby obtain the corresponding entity names and background knowledge. This differs from our two-stage approach in that it bypasses the secondary textual retrieval phase, which normally uses the entity name and input query to refine the knowledge selection. Instead, the single-stage method directly employs the retrieved entity background contexts as knowledge snippets for retrieval-augmented generation. We compare this single-stage retrieval method with our proposed two-stage retrieval method in Table 7. For a more comprehensive comparison, we also included RA-CM3 Yasunaga et al. (2023) for comparison as it employed a single-stage retrieval method.

## A.6 EFFECT OF THE NUMBER OF RETRIEVED KNOWLEDGE SNIPPETS

We investigate the impact of the number of textual knowledge snippets returned for each image during the second stage of retrieval, i.e., $l$ in Equ. 2, and show the results on the InfoSeek dataset in Table 8. LLaVA-v1.5 with 4 or 8 snippets denotes the LLaVA-v1.5 fine-tuned with retrieval augmentation but without visual token refinement and knowledge-intensive pertaining. As shown in the table, expanding the retrieval from top-4 to top-8 snippets results in marginal improvements, demonstrating the less sensitivity of our 2-stage retrieval strategy on the number of retrieved knowledge snippets.

## A.7 EFFECT OF TRUNCATION

We implement a truncation strategy for each retrieved knowledge snippet during tokenization to construct the multimodal interleaved input, preventing longer preceding retrieved knowledge snippets

Table 8: Performance comparison in accuracy (%) for VLMs with different numbers of retrieval knowledge snippets on the InfoSeek.

| Model | Entity | Query |
|---|---|---|
| LLaVA-v1.5 | | |
| - 4 snippets | 20.68 | 23.41 |
| - 8 snippets | 20.84 | 23.34 |
| RORA-VLM(ours) | | |
| - 4 snippets | 24.56 | 26.33 |
| - 8 snippets | 25.10 | 27.34 |

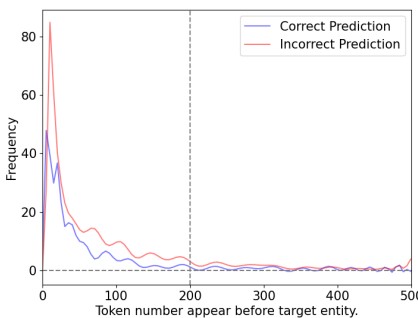

Table 9: Position distribution of the target entity name within retrieved knowledge snippets.

from dominating the limited input sequence space, thereby ensuring that subsequent retrieved information is preserved. However, this raises an important question: how much valuable information is lost due to this truncation?

To assess the potential loss of critical information, we examine instances where the retrieved knowledge snippets explicitly mention the target entity name. We count the number of tokens that appear before this mention and visualize the positional distribution of key information (i.e., the target entity name) within the retrieved snippets, as shown in Figure 9. As depicted, in most cases, the entity name appears within the first 200 tokens of the retrieved passages, whereas our truncation is applied at the 400-token mark for each passage. This buffer ensures a high retention rate of valuable information, minimizing the risk of discarding critical content due to truncation.

## A.8 DATASETS

**OVEN (Hu et al., 2023)**  OVEN is an entity recognition dataset constructed by repurposing 14 existing datasets, comprising over 5 million instances. All labels in OVEN are mapped onto a unified label space of Wikipedia entities. Each instance consists of an entity image paired with its corresponding entity name. The tasks in OVEN require vision-language models (VLMs) to recognize visual entities from a pool of six million possible Wikipedia entities.

**InfoSeek (Chen et al., 2023d)**  InfoSeek is a large-scale visual question answering (VQA) dataset focused on knowledge-seeking queries. It consists of over 1.35 million image-text pairs, each posing various questions about objects, scenes, and actions that require external knowledge—such as factual information—rather than solely relying on the visual content.

**Encyclopedic-VQA (Mensink et al., 2023)**  Encyclopedic-VQA is a knowledge-intensive VQA dataset containing over 221,000 image-text instances that require deep reasoning and access to external knowledge. It is well-suited for evaluating a model's ability to answer questions that extend beyond the image content.

## A.9 IMPLEMENTATION DETAILS

We adopt `LLaVA-v1.5-7B` (Liu et al., 2023a) as the backbone model for our RORA-VLM. In our experiments, limited by the input sequence length, we set the retrieval parameters as follows: $k = 3$ and $l = 3$ for image-anchored textual-query expansion, and $m = 144$ for our query-oriented visual token refinement method. All models are trained using 8 NVIDIA H100 GPUs. Both pre-training and fine-tuning processes follow the hyperparameters specified in the original LLaVA (Liu et al., 2023a) setup, ensuring consistency with previous work.

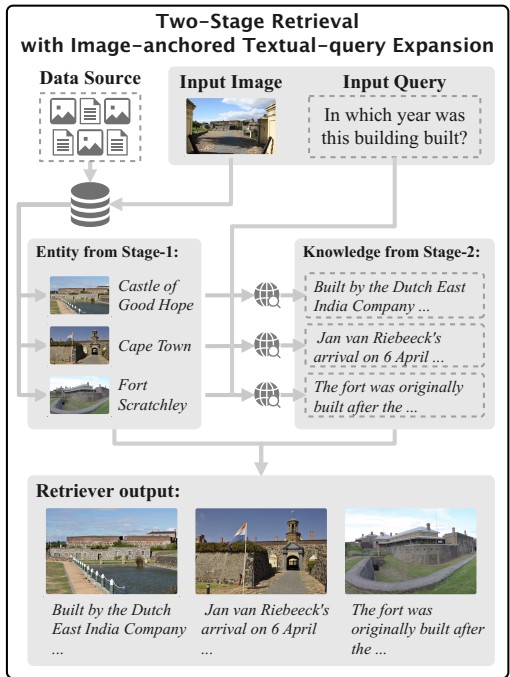

Figure 6: Overview of the Image-anchored Textual-query Expansion

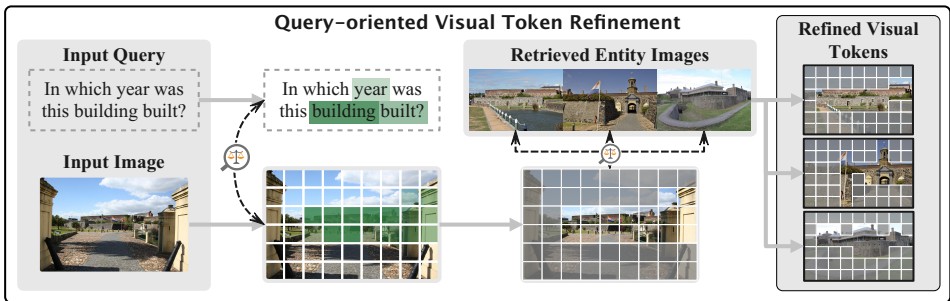

Figure 7: Overview of the Query-oriented Visual Token Refinement

## A.10    SCHEMATIC DIAGRAM OF THE 2-STAGE RETRIEVAL

We include Figure 6 to provide a more intuitive explanation of our proposed 2-stage retrieval.

## A.11    SCHEMATIC DIAGRAM OF THE QUERY-ORIENTED VISUAL TOKEN REFINEMENT

We include Figure 7 to provide a more intuitive explanation of the query-oriented visual token refinement.

