# OpenReview forum: "RoRA-VLM: Robust Retrieval-Augmented Vision Language Models"
_ICLR.cc/2025/Conference — Submitted to ICLR 2025_

### Official Review · Reviewer_ZXbA · 2024-11-02

**Soundness:** 3
**Presentation:** 3
**Contribution:** 3
**Rating:** 6
**Confidence:** 3

**Summary:**

The paper introduces RORA-VLM, a framework aimed at improving Vision-Language Models (VLMs) on knowledge-intensive tasks. The method addresses two challenges: (1) effectively retrieving relevant multimodal information given the inherent discrepancy between vision and language modalities, and (2) managing the noisy and extraneous information in retrieved knowledge. The paper’s contributions include a two-stage retrieval process with image-anchored textual-query expansion and a robust retrieval augmentation method that employs adversarial noise and visual token refinement. Extensive experiments demonstrate that RORA-VLM outperforms current models on benchmarks such as OVEN, InfoSeek, and Enc-VQA.

**Strengths:**

1. **Novelty**: The authors address two core challenges in multimodal retrieval-augmented generation (RAG): the retrieval inaccuracies caused by modality discrepancies and the noise often present in retrieved content. To tackle these issues, they propose an innovative solution using a two-stage retrieval process that mitigates modality inconsistency, allowing the system to capture multimodal background information more comprehensively. Combined with an anti-noise strategy, this approach effectively suppresses irrelevant information while enhancing retrieval accuracy and overall performance in multimodal tasks.

2. **Significance**: RORA-VLM offers a valuable method for improving VLMs, especially in knowledge-intensive domains, where retrieval-augmented tasks are often challenged by noise. This framework effectively addresses this key issue, making it particularly suitable for such applications.

3. **Clarity of Presentation**: The paper is well-structured with a clear research motivation, providing thorough explanations of the methodology and experimental results. This clarity aids readers in understanding both the approach and its effectiveness.

**Weaknesses:**

1. **Inconsistency in Method and Motivation**: The two-stage retrieval in the paper looks like an image entity-driven retrieval approach, would modal differences be better handled with image+query composite retrieval; Additionally, the motivations behind the designs of Query-oriented Visual Token Refinement and Adversarial Noise Injection for Robust Augmentation seem to conflict. The former introduces noise for adversarial learning, while the latter focuses on denoising. It might align better with the concept of adversarial learning if the former were applied solely during the inference phase and the latter exclusively during training.
2. **Fairness of Experimental Comparisons**: In the main experiments presented in Table 1, the authors' method has undergone pre-training and fine-tuning on knowledge-intensive datasets, whereas many baseline models may not have been trained on such datasets. This raises questions about the fairness of the experimental comparisons.
3. **Lack of Ablation Studies**: The paper lacks ablation studies on key parameters such as k, l, and m. Including these analyses would provide valuable insights into the impact of these parameters on the model's performance.

**Questions:**

1. **Differences in Approach and Motivation**: The two-stage retrieval approach proposed in the paper seems to be driven by image entities. Does a retrieval approach that combines images and queries better address pattern differences? Furthermore, how do the authors reconcile the conflicting motivations behind query-oriented visual token refinement (introducing noise for adversarial learning) and adversarial noise injection (focusing on denoising)? Would it be more consistent with adversarial learning principles if the former were used only during inference and the latter only during training?
2. **Fairness of experimental comparisons**: In Table 1, do the authors plan to conduct more experiments to ensure that all models are evaluated on a level playing field?
3. **Lack of ablation studies**: Can the authors provide insights on the impact of these parameters (k,l,m) on model performance?

---

> ### Author Response · Authors · 2024-11-23
>
> ## **W1. Comparing two-stage retrieval with image+query composite retrieval**
> We appreciate the thoughtful comment from the reviewer and humbly argue that our two-stage retrieval method is more flexible and potentially has better performance and efficiency. The reasons are as follows.
>
> During database construction, we only need to encode images as keys and their entity names as values. Once the main entity in an image is identified, we can simply combine the question with the entity name and search for relevant knowledge in a pure text database. In contrast, using an image+query embedding for searching requires constructing a search index that jointly represents both image content and knowledge, as seen in the knowledge base used in [1]. However, models capable of effectively encoding image+query pairs into embeddings are often not as powerful as models designed for generating embeddings within a single modality. Existing approaches often rely on ad hoc implementations, such as combining CLIP embeddings of images and text, which introduce many design questions and may result in suboptimal performance.
>
> In comparison, our proposed two-stage method is modular and can seamlessly integrate with any state-of-the-art image and text retriever, ensuring adaptability and robustness.
>
> To further demonstrate the advantages of our approach, we implemented two retrieval-augmented baselines that apply the multimodal composite retrieval, PreFLMR [2] and RA-CM3 [1], to empirically compare their performance with our proposed method. Both baselines were implemented using the same backbone models as our approach (Vicuna/LLaVA-1.5) to ensure a fair comparison. PreFLMR employs a multimodal retriever to retrieve fine-grained query-related textual contexts, while RA-CM3 encodes multimodal documents for mixed-modal retrieval.
>
> The experimental results, presented in the table below, demonstrate that while composite retrieval approaches such as RA-CM3 can benefit from richer cross-modal representations, they may struggle to match the adaptability and robustness of our modular two-stage retrieval approach in knowledge-intensive complex reasoning tasks.
>
> | Model           | InfoSeek - Entity | InfoSeek - Query |
> |-----------------|-------------------|------------------|
> | LLaVA-v1.5      |       10.34       |       12.98      |
> | PreFLMR         |       19.37       |       22.21      |
> | RA-CM3          |       17.09       |       21.64      |
> | RoRA-VLM (ours) |       **25.10**       |       **27.34**      |
>
> Table: Evaluation results in accuracy (%). The best performance is highlighted in **bold**.
>
> ## **W2. Confliction between visual token refinement and adversarial noise injection**
> We appreciate the reviewer’s feedback and would like to clarify that the motivations behind visual token refinement and adversarial noise injection are distinct but complementary, working toward the same overarching goal.
>
> Adversarial noise injection during training is designed to help the model effectively leverage visual modality information by comparing retrieved entity images to the query image. This process enables the model to identify query-relevant retrieved documents while filtering out irrelevant retrieval passage noise. Its primary focus is on improving the model's robustness against noisy or irrelevant information in the retrieved documents.
>
> On the other hand, visual token refinement is specifically tailored to support the adversarial noise injection training process. It filters out query-irrelevant content from the input images, ensuring that only query-relevant visual information is retained. By removing distracting elements such as background content or query-irrelevant entities, visual token refinement facilitates more accurate comparisons of entity-level content between images. This design ensures that the model is less influenced by extraneous visual information, improving its ability to focus on the content most relevant to the query.
>
> Thus, while the two components address different aspects of the problem—visual token refinement focuses on image-level filtering, and adversarial noise injection focuses on retrieval-level robustness—they are aligned in their purpose of enhancing the model's ability to handle noise.

---

> ### Author Response · Authors · 2024-11-23
>
> ## **W3. Fairness of Experiment Comparison in Table 1**
> We are sorry for missing the training details of the baselines. To clarify, all the baselines in Table 1 are finetuned on OVEN, InfoSeek and Enc-VQA, respectively, so that we can ensure a fair comparison between our approach and all the baselines
>
> We list the knowledge-intensive pretraining as one of our contributions since previous vision-language models were predominantly pretrained on image-caption datasets such as CC12M. Pretraining solely on image-caption pairs may not be sufficient to align the internal knowledge in large language models (LLMs), such as entity names and entity background knowledge, with the visual representations of entities in images. In Table 4, we perform ablation studies to show the effect of knowledge-intensive pretraining on both LLaVA-v1.5 and our approach: LLaVA-v1.5 v.s. LLaVA-v1.5 w/ WikiWeb2M, RORA-VLM v.s. RORA-VLM w/o WikiWeb2M. Our results indicate that pretraining on image and entity-rich captions can significantly improve VLMs’ performance on information-seeking tasks, highlighting an important direction for advancing VLMs in the future.
>
> ## **W4. Ablation study on hyper-parameters**
> We appreciate the reviewer’s comment regarding the impact of the parameters k, l, and m on model performance. Due to the time constraints of the rebuttal period, we were unable to conduct ablation studies for all these parameters. However, recognizing the importance of these analyses, we have already included an ablation study on the most critical hyperparameter, m (the number of retrieved knowledge snippets), in Appendix A2 of our submission. This study provides insights into how varying m affects model performance.
>
> For the other suggested ablation studies, we acknowledge their value and will ensure they are included in the revised version of the paper if our submission is accepted.
>
> ## **References**
> [1] Michihiro Yasunaga, Armen Aghajanyan, Weijia Shi, Rich James, Jure Leskovec, Percy Liang, Mike Lewis, Luke Zettlemoyer, and Wen-tau Yih. 2023. Retrieval-augmented multimodal language modeling. In Proceedings of the 40th International Conference on Machine Learning (ICML'23), Vol. 202. JMLR.org, Article 1659, 39755–39769.
>
> [2] Weizhe Lin, Jingbiao Mei, Jinghong Chen, and Bill Byrne. 2024. PreFLMR: Scaling Up Fine-Grained Late-Interaction Multi-modal Retrievers. In Proceedings of the 62nd Annual Meeting of the Association for Computational Linguistics (Volume 1: Long Papers), pages 5294–5316, Bangkok, Thailand. Association for Computational Linguistics.

---

> > ### Author Response · Authors · 2024-11-25
> >
> > Dear Reviewer ZXbA,
> >
> > We sincerely appreciate the time and effort you’ve devoted to reviewing our work. We understand that your schedule may be quite busy, and we are truly grateful for your valuable feedback. As the Author-Reviewer discussion phase is ending soon, we would greatly value the opportunity to engage in further discussion with you. Our aim is to gain insights into whether our responses effectively address your concerns and to ascertain if there are any additional questions or points you would like to discuss.
> > We look forward to the opportunity for further discussion with you. Thank you for your thoughtful consideration.
> >
> > Best regards,
> >
> > Authors

---

> ### Comment · Reviewer_ZXbA · 2024-11-27
>
> I sincerely thank the authors for their detailed rebuttal, which has addressed most of my concerns. I decide to maintain my positive score. Here are some additional comments:
>
> Regarding W1:
>
> Thank you for the detailed explanation and additional experiments. Based on the authors’ response, I understand that multimodal composite retrieval excels in capturing multimodal semantics, while the entity-driven method shows superior performance in entity-centric, knowledge-intensive, complex reasoning tasks. Including explicit clarifications and comparisons in the paper would be beneficial to elucidate the trade-offs between these two approaches.
>
> Regarding W2:
>
> I appreciate the authors’ clarification. However, the statement that “visual token refinement is specifically tailored to support the adversarial noise injection training process” remains somewhat unclear. My understanding is that the two modules are not tightly coupled. Visual token refinement aims at denoising, while noise injection introduces adversarial noise for robust training. Using both during training might potentially interfere with the adversarial learning dynamics. It could be insightful to validate this hypothesis by employing noise injection during training and incorporating visual token refinement exclusively during inference. This additional experiment might help clarify their interplay.

---

> > ### Author Response · Authors · 2024-12-03
> > **Thank you for your response**
> >
> > We sincerely appreciate your time and effort in reviewing our paper and providing valuable feedback, which is crucial for improving our work. We are glad that our responses have addressed your concerns. In the revised manuscript, we will incorporate the detailed explanations and comparisons as discussed, as well as include the additional denoising experiment you suggested.

---

> > ### Author Response · Authors · 2024-12-03
> >
> > We appreciate the reviewer's feedback and suggestions. We would like to further clarify the relationship between the two modules and address the concerns raised.
> >
> > While it is correct that the visual token refinement module and the adversarial noise injection training process are not tightly coupled (i.e., the former is not a necessary condition for the latter), the adversarial noise injection process does benefit from the visual token refinement module, rather than conflicting with it. Specifically, the types of noise addressed by each module are fundamentally different. The visual token refinement process focuses on removing query-irrelevant visual entities or objects within the input images. Even when the retrieved images are query-relevant entity images, they may contain background content or other distracting elements unrelated to the query. By eliminating these distractions, visual token refinement ensures that the model's attention is directed solely toward query-relevant visual information. Conversely, adversarial noise injection addresses a different challenge: the noise introduced by unsuccessful retrievals. The goal of adversarial training is to enhance the model's robustness against irrelevant retrieval passages by encouraging it to better compare the query entity image with retrieved entity images. By refining the input images through visual token refinement, the adversarial training process can focus more effectively on entity-level comparisons.
> >
> > Following the reviewer's suggestion, we conducted an additional ablation experiment where visual token refinement was applied exclusively during inference without being included in the training phase. The results are presented in the table below:
> >
> > | Model                       | InfoSeek - Entity | InfoSeek - Query |
> > |-----------------------------|-------------------|------------------|
> > | LLaVA-v1.5                 |       10.34       |       12.98      |
> > | w/o VK-Refinement (during Training) |       22.12       |       24.42      |
> > | RoRA-VLM (ours)             |    **25.10**      |    **27.34**     |
> >
> > Table: Evaluation results in accuracy (%). The best performance is highlighted in **bold**.
> >
> > The results indicate that excluding visual token refinement during training leads to performance degradation. This outcome is expected as the refinement process alters the visual embedding arrangement of the original CLIP encoder. Using visual token refinement solely at inference time without incorporating it during training could lead to performance degradation.
> >
> > Based on this additional ablation study, we conclude that visual token refinement and adversarial noise injection are not in conflict but rather complement each other. Together, they address different aspects of the problem and contribute to the overall robustness and performance of the model.

---

### Official Review · Reviewer_MWJ4 · 2024-11-03

**Soundness:** 3
**Presentation:** 2
**Contribution:** 2
**Rating:** 5
**Confidence:** 3

**Summary:**

This paper presents a multimodal version RAG targeting multimodal large language models, such as LLaVA-1.5 for information-seeking VQA. To solve two challenges, the authors propose a 2-stage retrieval process with image-anchored textual query expansion and noise-resilient retrieval augmented generation. Experimental results highlight its effectiveness on OVEN and InfoSeek benchmarks.

**Strengths:**

1. The motivation for this work is clearly presented and easy to follow.
2. Experiment results demonstrate its effectiveness.

**Weaknesses:**

1. In the Introduction section, providing a figure showing the process of the 2-stage retrieval method would be easier to understand.
2. Related work discussion and method novelty. How to incorporate multi-modal knowledge into models is not a new problem[1][2]. Some related work is proposed in other multi-modal tasks, such as knowledge-based VQA. Besides, adversarial training is also adopted in existing vision and language training, such as [3]. The authors are encouraged to discuss the existing work and compare the related ones with the proposed method.
3. The correspondence between the ablation model variants in Table 2 and the proposed module is somewhat unclear. What about the ablation of the two-stage retrieval ?
4. Figure 2 lacks some of the details of the methodology. The authors are encouraged to refine it.

[1] Gui, Liangke, et al. "KAT: A Knowledge Augmented Transformer for Vision-and-Language." Proceedings of the 2022 Conference of the North American Chapter of the Association for Computational Linguistics: Human Language Technologies. 2022.
[2] Lin, Weizhe, et al. "Fine-grained late-interaction multi-modal retrieval for retrieval augmented visual question answering." Advances in Neural Information Processing Systems 36 (2023): 22820-22840.
[3] Gan, Zhe, et al. "Large-scale adversarial training for vision-and-language representation learning." Advances in Neural Information Processing Systems 33 (2020): 6616-6628.

**Questions:**

Please refer to the above section.

---

> ### Author Response · Authors · 2024-11-23
>
> ## **W1. Needs a figure to show 2-stage retrieval**
> We appreciate the reviewer’s suggestion to include a figure illustrating the two-stage retrieval method in the Introduction section. We agree that a detailed visual representation would enhance the clarity and understanding of the proposed approach. Accordingly, we will add a detailed figure to illustrate the two-stage retrieval process in the revised version.
>
> ## **W2. Discussion of related studies and novelty of our method.**
> We provide the following discussion to highlight the distinctions and contributions of our method, which will also be included in the revised version of the draft.
>
> Previous studies, such as [1], [2], and the more recent work [4], focus on multimodal retrieval methods that enhance text-only language models by retrieving textual knowledge using visual queries (e.g., an image) to assist in answering visual questions. These approaches primarily aim to improve retrieval quality to support downstream tasks better.
>
> In contrast, our work focuses more on addressing the critical challenge of how to more effectively utilize the retrieved information in retrieval-augmented generation (RAG). While prior work [1][2] and recent models, such as RA-CM3 [5] and Wiki LLaVA, largely rely on the quality of the retrieved passages, they do not explicitly account for the inherent noise and irrelevance introduced by multimodal retrieval processes. Given that the recall@1 of state-of-the-art retrievers on datasets like InfoSeek is still below 0.2, the presence of noisy or irrelevant passages remains a significant limitation for RAG systems. Our method, RoRA-VLM, is the first to directly tackle this issue by proposing a robust solution to reduce retrieval-induced noise. Unlike previous approaches that rely exclusively on textual retrieved information, our framework fully utilizes multiple modalities of retrieved information during the generation process. By enabling the model to learn to distinguish relevant information from irrelevant noise within the retrieved materials, our approach significantly enhances the robustness of the RAG pipeline. This allows the vision-language models to maintain strong performance even when retrieval accuracy is imperfect.
>
> [3] primarily focuses on visual representation learning, where noise is introduced at the embedding level of images and text. In contrast, our method focuses on injecting noises into the retrieved knowledge for generative models. The noise injection process and the corresponding learning paradigm in our approach are fundamentally different from those in [3].
>
> ## **W3. Correspondence between the ablation model variants in Table 2 and the proposed module.**
>
> The first row in Table 2 presents the performance of our RoRA-VLM model on InfoSeek. The second row, labeled as RoRA-VLM w/o VK-Refinement, evaluates the impact of our proposed visual token refinement strategy. Specifically, in RoRA-VLM w/o VK-Refinement, all configurations remain the same as in RoRA-VLM, except that the visual token refinement method is not applied to the input image tokens. Since the sequence length of LLaVA is limited to 2048, to accommodate four images and the textual context, we reduce the number of image tokens for each image by employing 2D average pooling, to reduce the number of input image tokens from 576 to 144 and match the number of visual tokens used in our VK-Refinement method.
>
> In the third row, we assess the effectiveness of Adversarial Noise Injection for Robust Augmentation. To demonstrate that the performance improvements of our model come from its ability to distinguish relevant knowledge (associated with the entity in query image) from irrelevant knowledge or noise—and not simply from the availability of additional retrieved knowledge—we conduct an ablation study. In this ablation, we keep the same two-stage retrieval process but only provide the retrieved textual knowledge to the VLM during both training and inference, omitting the retrieved images entirely, so that the model relies solely on textual information to make predictions and cannot leverage visual information as evidence to validate the correctness of retrieval. This variant, labeled RoRA-VLM text-only RAG, ensures that the VLM processes the same textual knowledge as in the proposed approach but without the additional image input. As shown in Table 2, this results in a significant performance drop, which demonstrates that the improvements achieved by our adversarial training stem from the model's ability to effectively filter and focus on relevant knowledge rather than simply benefiting from the additional knowledge.

---

> ### Author Response · Authors · 2024-11-23
>
> ## **W4. Ablation of the two-stage retrieval**
> Following the reviewer’s suggestion, we conducted an additional ablation study to emphasize the effectiveness of our two-stage retrieval approach. Specifically, we performed an ablation experiment using only a single-stage retrieval method. In the single-stage configuration, we utilized the CLIP embedding of the query image to retrieve the most similar entity images in our retrieval database, and thereby obtain the corresponding entity names and background knowledge. This differs from our two-stage approach in that it bypasses the secondary textual retrieval phase, which normally uses the entity name and input query to refine the knowledge selection. Instead, the single-stage method directly employs the retrieved entity background contexts as knowledge snippets for retrieval-augmented generation. We compare this single-stage retrieval method with our proposed two-stage retrieval method in the table below. For a more comprehensive comparison, we also included RA-CM3 for comparison as it employed a single-stage retrieval method.
>
> | Model                   | InfoSeek - Entity | InfoSeek - Query |
> |-------------------------|-------------------|------------------|
> | LLaVA-v1.5              |       10.34       |       12.98      |
> | RA-CM3 (single-stage)   |       17.09       |       21.64      |
> | RoRA-VLM (single-stage) |        21.9       |       23.87      |
> | RoRA-VLM (2-stage)      |      **25.10**       |       **27.34**      |
>
> Table: Evaluation results in accuracy (%). The best performance is highlighted in **bold**.
>
> From the results, it is evident that our two-stage retrieval method outperforms the single-stage approaches. A likely reason for this superiority is the flexibility and efficiency of our two-stage method. Specifically, during database construction, we only encode images as keys and their corresponding entity names as values. Once the main entity in an image is identified, we can combine the query with the entity name and efficiently search for relevant knowledge in a purely textual database.
>
> In contrast, single-stage retrieval methods require constructing a search index that jointly represents both image content and knowledge, as seen in knowledge bases like [1]. However, models capable of effectively encoding image-query pairs into embeddings often underperform compared to models optimized for embedding generation within a single modality. Existing approaches typically rely on ad hoc implementations, such as combining CLIP embeddings of images and text. These methods introduce various design challenges and can lead to suboptimal performance.
>
> In comparison, our proposed two-stage retrieval method is modular and can seamlessly integrate with state-of-the-art image and text retrievers, ensuring greater adaptability and robustness. We hope this study further demonstrates the advantages of our approach and addresses the reviewer’s concerns.
>
> ## **W5. Refine Figure 2 to include more details of the methodology**
> We appreciate the reviewer’s comments and suggestions. We agree that adding visual representations would greatly improve the clarity and understanding of our approach. In the revised version, we will include two more figures: one to illustrate the details of the two-stage retrieval process and another to depict the query-oriented visual token refinement process.
>
> ## **References**
> [1] Gui, Liangke, et al. "KAT: A Knowledge Augmented Transformer for Vision-and-Language." Proceedings of the 2022 Conference of the North American Chapter of the Association for Computational Linguistics: Human Language Technologies. 2022.
>
> [2] Lin, Weizhe, et al. "Fine-grained late-interaction multi-modal retrieval for retrieval augmented visual question answering." Advances in Neural Information Processing Systems 36 (2023): 22820-22840.
>
> [3] Gan, Zhe, et al. "Large-scale adversarial training for vision-and-language representation learning." Advances in Neural Information Processing Systems 33 (2020): 6616-6628.
>
> [4] PreFLMR: Scaling Up Fine-Grained Late-Interaction Multi-modal Retrievers. Weizhe Lin, Jingbiao Mei, Jinghong Chen, Bill Byrne
>
> [5] Michihiro Yasunaga, Armen Aghajanyan, Weijia Shi, Rich James, Jure Leskovec, Percy Liang, Mike Lewis, Luke Zettlemoyer, and Wen-tau Yih. 2023. Retrieval-augmented multimodal language modeling. In Proceedings of the 40th International Conference on Machine Learning (ICML'23), Vol. 202. JMLR.org, Article 1659, 39755–39769.

---

> > ### Author Response · Authors · 2024-11-25
> >
> > Dear Reviewer MWJ4,
> >
> > We sincerely appreciate the time and effort you’ve devoted to reviewing our work. We understand that your schedule may be quite busy, and we are truly grateful for your valuable feedback. As the Author-Reviewer discussion phase is ending soon, we would greatly value the opportunity to engage in further discussion with you. Our aim is to gain insights into whether our responses effectively address your concerns and to ascertain if there are any additional questions or points you would like to discuss.
> > We look forward to the opportunity for further discussion with you. Thank you for your thoughtful consideration.
> >
> > Best regards,
> >
> > Authors

---

> > > ### Author Response · Authors · 2024-11-28
> > >
> > > Dear Reviewer MWJ4,
> > >
> > > We have posted the revised version of our paper, incorporating all additional details and experiments mentioned during the rebuttal period, highlighted in blue for your convenience.
> > >
> > > As the deadline approaches, we kindly ask you to review the revisions and reevaluate our work based on these updates. Please let us know if you have any further questions or need clarification. We sincerely appreciate your time and effort in reviewing our work.
> > >
> > > Best regards,
> > >
> > > Authors

---

### Official Review · Reviewer_XFpf · 2024-11-03

**Soundness:** 3
**Presentation:** 3
**Contribution:** 3
**Rating:** 6
**Confidence:** 4

**Summary:**

The paper introduces RORA-VLM, a retrieval-augmented framework designed to enhance Vision-Language Models (VLMs) by addressing two main challenges: managing multimodal query discrepancies and filtering out irrelevant, noisy retrievals. RORA-VLM employs a two-stage retrieval process: 1) Image-Anchored Entity Retrieval: This stage retrieves visually similar images based on the query image, anchoring the retrieval with associated entity information 2) Query-Expanded Text Retrieval: Using entity names from the first stage, the method expands the query to retrieve additional textual knowledge from sources like Google Search.

**Strengths:**

- The paper is well written and easy to follow
- The authors clearly state the motivation for the proposed method and its necessity.
- RORA-VLM introduces a unique two-stage retrieval approach, effectively bridging the gap between visual and textual information for more accurate knowledge retrieval.
- The paper tackles the common issue of irrelevant or noisy data in retrieval-based methods by implementing noise resilience strategy
- The paper address a clearly practical application that might be useful for the community and the industry.

**Weaknesses:**

Method:

- Section 3.2: The authors describe in details two stages. For my understanding, stage-1 is just formulation of the K-NN of the query image in the WIT images (within CLIP latent space). This is a well-known concept, especially in this line of work. I think this is well-detailed stage but it should be on the appendix, while the main paper should contain a brief description of the stage.
- Line 270: “Similarly, the image I is encoded into a sequence of visual embeddings…” - this is not clear. CLIP encodes an image/text intro a shared embeddings space of dimension d. How do you encode the image patches (n) to the same dimension? Do you feed-forward each patch, separately, to the CLIP model? Do you use the N internal CLIP features for each patch? If so, are you sure that their dimension is d, before the last projection layer? Do you project them with the last visual projection layer as the pooled [CLS] token projected? Please elaborate more on this procedure.

Section 5 currently combines results, ablation study, and discussion, which affects the clarity and flow of these findings. Separating these into distinct sections—such as “Results,” “Ablation Study,” and “Discussion”—would make it easier for readers to follow each component and understand the contributions more clearly. Additionally, crucial details and experiments appear to be missing, and some existing experiments do not convincingly support the claims made. Below are specific areas where the section could be strengthened:

Evaluation:

- Main results: Lines 307-316 (Baselines): The authors list a several MLLM backbones for the QA task which is great. However,  baselines to compare to should be other RAG methods. If I understand correctly, only RORA-VLM and Wiki-LLaVA* are using Retrieval Augmentations. If so, how is it comparable to other baselines that uses zero-shot?
- Building on previous point, I am not fully understand the entire training setup: the only model that was tuned (lines 317-345) was RORA-VLM? If so, again, how is it comparable to other baselines? Please clarify these points.

There are not enough details about the evaluation protocols and datasets in the paper, and some comparisons are missing. For example, what was the training set of each baseline in Table 1? Did the authors fine-tuned each baseline on the same dataset? which one of them use the proposed RAG method? what about other RAG methods and baselines?

Ablation Study:

- Lines 365-367 states “we use the widely adopted average pooling (kernel size of 2, stride of 2) to obtain the same number of visual tokens as our refinement approach” What does “widely adopted average pooling” mean on N CLIP vectors? How does it relate to a kernel size of size 2 and stride 2? Did you manipulated the input image/kernel of CLIP to get the same amount of CLIP vectors? The authors should elaborate on the experiment that was done here, it is unclear.
- Lines 405-419:  I am not convinced why this experiment proves that the model ignore “noise” in the retrieval samples. I would be more convinced with an following experiments, for example: providing the model 1 relevant sample with 2 other randomly-sampled ones, will not change the model’s answer, regardless which 2 noise samples were chosen, or by just proving the 1 relevant sample with no other samples.
- Lines 420-430 describe Figure 4 that supposed to show how the model ignore “noise” samples. However, it seems like the model pays attention to specific words the correlate with the question (e.g., row 1,  “how wide…” attend “height” and “width”). These examples does not show any rubsness to “noise” retrieval as intended.

**Questions:**

- In line 257 - do you mean top-2 (instead of top-(k-1))?

---

> ### Author Response · Authors · 2024-11-23
>
> ## **W1. Shorten the description of Stage-1.**
>
> We appreciate the suggestion from the reviewer and admit that Stage-1 is quite similar to the KNN formulation illustrated by the reviewer. However, besides providing the details of the technical design, the discussion of Stage-1 is more about providing the setups of the multimodal-based retrieval augmentation process, such as the source of the retrieval augmentation (i.e., an image database built on 37.6M images from WIT) and the choice of image encoder (CLIP), which are different from relevant previous studies and necessary to help readers gain a better understanding of the problem setup. The following sections also refer to some variable names defined at this stage.  Following the reviewer's suggestions, we will shorten the description and move some details to the appendix in the revised version.
>
> ## **W2. How to encode an image into a sequence of visual embeddings using CLIP? do you encode the image patches (n) to the same dimension? Do you feed-forward each patch, separately, to the CLIP model? Do you use the N internal CLIP features for each patch? If so, are you sure that their dimension is d, before the last projection layer? Do you project them with the last visual projection layer as the pooled [CLS] token projected? Please elaborate more on this procedure.**
>
> Below, we provide a detailed description of how we encode an image into a sequence of visual embeddings using CLIP, addressing each aspect of the reviewer's concerns.
>
> **Image Encoding with CLIP:** In the CLIP model, the visual encoder is based on the Vision Transformer (ViT) architecture. Given an image, the visual encoder processes it as a whole and encodes it into a feature representation of shape [576, 1024]. This representation can be interpreted as 576 vectors, each with a dimensionality of 1024. The 576 vectors correspond to patches of the input image, where the image is internally divided into a grid of patches during the encoding process. This division is not explicit; rather, it is an inherent part of the ViT architecture, which computes patch-level embeddings directly through a convolutional embedding layer applied to the full image. The resulting intermediate patch embeddings collectively form the image’s representation in the model’s latent space.
>
> **Dimensionality of Visual Embeddings:** After passing through the vision transformer (ViT) layers, each patch is represented as a feature vector with a dimensionality of 1024. To further process these features, we utilized the final visual projection layer of the original CLIP model. This projection layer, which is also used for the pooled [CLS] token in the original implementation, is applied to all 576 patch-based feature vectors in our approach. The projection reduces the dimensionality of each feature vector from 1024 to 768. To clarify further, the visual projection layer is part of CLIP’s original implementation. While it is typically applied only to the pooled [CLS] token to produce the image-level feature representation, in our work, we extend its application to all 576 patch-level feature vectors. As a result, the output is a feature representation of shape [576, 768], where 576 corresponds to the number of patches and 768 is the dimensionality of the projected patch embeddings.
>
> After computing the patch embeddings, for each text query, we derive a 768-dimensional vector from the [CLS] token of the CLIP text encoder. We then compute the similarities between the text embedding and the image patch embeddings to select the top-m relevant patches, which are subsequently projected into the LLM's latent space using the LLaVA projector.

---

> > ### Author Response · Authors · 2024-11-23
> >
> > ## **W3. Lacks of retrieval augmented baselines.**
> >
> > We appreciate the comment from the reviewer and implemented two additional retrieval-augmented baselines [1][2]. Both baselines were implemented with the same backbone models as our approach (i.e., Vicuna/LLaVA-1.5 as the backbone model) to ensure a fair comparison.
> > - Baseline 1 – PreFLMR: it employs a multimodal retriever to retrieve query-related fine-grained textual context, which is then used to support the language model in answering questions. ​​PreFLMR relies on its own constructed database, which includes the WIT dataset and other sources. Therefore, it has access to a knowledge base richer than our model.
> > - Baseline 2 – RA-CM3: it encodes multimodal documents for mixed-modal retrieval. The retrieved multimodal documents are subsequently fed into a multimodal model to augment the generation process. As the source code for this baseline was not publicly available at the time of submission, we reimplemented it based on our best understanding of the paper descriptions. We constructed the retriever using the same data source (the WIT dataset) that we employed for our model's retriever. This means both our model and Baseline 2 retrieve information from the same dataset.
> >
> > Given the limited time window for rebuttal, we could only try our best to fine-tune and evaluate these additional baselines on the InfoSeek dataset, which is the most challenging and comprehensive dataset used in our evaluation (all ablation studies were also conducted on this dataset). We will continue the experiments on other evaluation datasets in our paper draft and report the results in the next revised version. The table below shows their quantitative results.
> >
> > |    Model    | InfoSeek - Entity       |  InfoSeek - Query |
> > |-------------------|:--------------:|:-----:|
> > | LLaVA-v1.5        | 10.34          | 12.98 |
> > | PreFLMR           | 19.37          | 22.21 |
> > | RA-CM3            | 17.09          | 21.64 |
> > | RoRA-VLM (ours)   | **25.10**          | **27.34** |
> >
> > Table: Evaluation results in accuracy (%). The best performance is highlighted in **bold**.
> >
> > From the results in Table 1, we observe that our proposed method outperforms all baseline models, demonstrating the effectiveness of the RORA framework. A possible reason for this is that the baseline methods do not explicitly address the noise inherent in the multimodal retrieval process. This limitation is significant, as the recall@1 of state-of-the-art retrievers (e.g., PreFLMR) on the InfoSeek dataset is currently below 0.2. This indicates that, in most cases, the retrieved knowledge snippets contain substantial noise. On the other hand, our RORA-VLM framework introduces a novel solution to mitigate retrieval-induced noise, thereby enhancing the model's robustness and overall performance.
> >
> > ## **W4. Training details for baselines.**
> > We are sorry for missing the training details of the baselines. To clarify, all the baselines in Table 1 are finetuned on OVEN, InfoSeek and Enc-VQA, respectively, so that we can ensure a fair comparison between our approach and all the baselines.
> >
> > ## **W5. Average pooling on CLIP vectors in Ablation Study (Lines 365-367).**
> > As detailed in the response for W2, each image is processed into a feature matrix with shape [576, 768] by the CLIP visual encoder and the LLaVA projector. Our proposed Visual Token Refinement method further selects the top 144 visual tokens that are most relevant to the query, constructing a feature matrix of shape [144, 768]. This selection process enables the VLM to focus more effectively on query-relevant image content while mitigating the influence of irrelevant noise, such as image backgrounds or query-irrelevant entities present in the image.
> >
> > To conduct an ablation study of the Visual Token Refinement method, we replace it with a simple average-pooling-based baseline, which also takes in the original [576, 768] visual patch vectors as input, downsample and convert them into [144, 768] vectors to ensure a fair comparison with our Visual Token Refinement method. Specifically, we first reshape the first dimension of the feature matrix (i.e., 576) into a 2D grid with dimensions 24 × 24, corresponding to the spatial arrangement of patches in the original image, then apply a 2D average pooling operation with a kernel size of 2 × 2 and a stride of 2. This pooling reduces the spatial resolution from 24 × 24 to 12 × 12, yielding 144 patch vectors in total while each patch vector has a dimensionality of 768.
> >
> > By reducing the number of feature vectors from 576 to 144, this process ensures compatibility with the limited sequence length of the LLM and aligns the number of input tokens for the average pooling baseline with that of our visual token refinement method. This alignment allows for a direct and fair comparison of the two approaches in the ablation study.

---

> > > ### Author Response · Authors · 2024-11-23
> > >
> > > ## **W6. More experiments to prove that the model ignores “noise” in the retrieval samples (Lines 405-419).**
> > > We appreciate the constructive comment and suggestion from the reviewer and address the concern with the following additional experiments.
> > >
> > > The key challenge for ideally proving the effectiveness of our model in ignoring the retrieval noise is that, for all the evaluation datasets we used, there are no gold standard labels for the retrieval step (i.e., we don’t know the exact relevancy between each input query and all the candidate samples for retrieval), so we cannot set up the experiment with 1 relevant sample and 2 randomly sampled irrelevant samples.
> > >
> > > However, we designed a similar experiment with varying levels of retrieval noise: During the inference stage, instead of using the top-3 retrieved entity images and their corresponding knowledge snippets, we used the top-1 retrieved entity image and knowledge snippet along with 2 randomly sampled irrelevant entity images and knowledge snippets. This sampling was repeated twice, yielding two different sets of randomly sampled irrelevant entity images and knowledge snippets for the same input instance. Then, based on the 3 sets of retrieved entity images and knowledge snippets, we perform retrieval augmentation on the InfoSeek dataset. The results are presented in the following table:
> > >
> > > | Model                          | InfoSeek - Entity | InfoSeek - Query |
> > > |--------------------------------|-------------------|------------------|
> > > | Top-3 Retreival                |       25.10       |       27.34      |
> > > | Top-1 Retreival + 2 Noises (1) |       19.61       |       21.97      |
> > > | Top-1 Retreival + 2 Noises (2) |       19.63       |       22.02      |
> > >
> > > Table: Evaluation results in accuracy (%).
> > >
> > > From the results, we observe that the model's performance remains unaffected regardless of which two noise samples were chosen, which to some extend proves the effectiveness of our model in identifying the useful information from the retrieved samples, regardless of the irrelevant samples. However, since we do not have ground-truth labels for the retrieval process, there is no assurance that the top-1 retrieval output is correct or not.  Therefore, it is reasonable to observe a slight performance degradation when randomly sampled irrelevant entities are used to replace the top-2 and top-3 retrieved samples. We hope these additional experiments can demonstrate the robustness of our method to retrieval noise.
> > >
> > > ## **W7. Clarification for Figure 4 (Lines 420-430).**
> > > We would like to clarify that the goal of Figure 4 is to show that by comparing the detailed visual content in the query image and the retrieved images, the model can identify which retrieved images contain the same visual entities as the query image. Once these relevant images are identified, the model focuses more on the textual knowledge associated with them. For example, in the middle row of Figure 4, the attention-score graph in the right column demonstrates that the model assigns higher attention to the text associated with the first image, while giving less attention to the text associated with the second and third images. This clearly indicates that through our adversarial training, the model learns to distinguish relevant knowledge from irrelevant information by referencing visual information, and hence becomes more robust to retrieved noises.
> > >
> > > ## **Q1. In line 257 - do you mean top-2 (instead of top-(k-1))?**
> > > Yes, in our specific setup, k = 3 and k-1 = 2.
> > >
> > > ## **References**
> > >
> > > [1] Weizhe Lin, Jingbiao Mei, Jinghong Chen, and Bill Byrne. 2024. PreFLMR: Scaling Up Fine-Grained Late-Interaction Multi-modal Retrievers. In Proceedings of the 62nd Annual Meeting of the Association for Computational Linguistics (Volume 1: Long Papers), pages 5294–5316, Bangkok, Thailand. Association for Computational Linguistics.
> > >
> > > [2] Michihiro Yasunaga, Armen Aghajanyan, Weijia Shi, Rich James, Jure Leskovec, Percy Liang, Mike Lewis, Luke Zettlemoyer, and Wen-tau Yih. 2023. Retrieval-augmented multimodal language modeling. In Proceedings of the 40th International Conference on Machine Learning (ICML'23), Vol. 202. JMLR.org, Article 1659, 39755–39769.

---

> > > > ### Comment · Reviewer_XFpf · 2024-11-24
> > > >
> > > > I thank the authors for detailed rebuttal and the effort they invested in preparing it.
> > > >
> > > > Regarding W4: Just to confirm my understanding, was each model trained separately on each of these datasets and then evaluated on the same dataset it was trained on? Is that correct?
> > > >
> > > > Answer to W3: This was a well-conducted experiment. The remaining step to complete it would be to evaluate the model’s performance using only the top-1 retrievals. This would demonstrate that adding two random samples does not impact the performance, whereas including two “relevant” (model-selected) retrievals leads to a performance improvement.
> > > >
> > > > I want to emphasize that all the details and experiments provided in this rebuttal are critical and should be included in the revised version of the paper, as my rating is largely based on their absence. Please ensure that these changes are marked in BLUE before submission. If space limitations are an issue, including them in the appendix with appropriate references in the main text would be acceptable.

---

> > > > > ### Author Response · Authors · 2024-11-24
> > > > >
> > > > > ## **W4: Regarding W4: Just to confirm my understanding, was each model trained separately on each of these datasets and then evaluated on the same dataset it was trained on? Is that correct?**
> > > > > Yes, the reviewer’s understanding is correct. All the baseline models and our proposed models were fine-tuned separately on the OVEN, InfoSeek, and Enc-VQA datasets, and then evaluated on the same dataset they were fine-tuned on. The evaluation on the three datasets was conducted independently, ensuring no overlap or interdependence between the evaluations.
> > > > >
> > > > > ## **W3: The remaining step to complete it would be to evaluate the model’s performance using only the top-1 retrievals.**
> > > > > Thank you for the reviewer’s valuable suggestion regarding the remaining step. Following the reviewer’s suggestion, we conducted the additional experiment, performing an ablation study using only the top-1 retrieved entity image and its corresponding knowledge snippet to augment the generation. The results are presented in the table below:
> > > > >
> > > > > | Model                          | InfoSeek - Entity | InfoSeek - Query |
> > > > > |--------------------------------|-------------------|------------------|
> > > > > | Top-3 Retrieval                |     **25.10**     |     **27.34**    |
> > > > > | Top-1 Retrieval + 2 Noises (1) |       19.61       |       21.97      |
> > > > > | Top-1 Retrieval + 2 Noises (2) |       19.63       |       22.02      |
> > > > > | Top-1 Retrieval Only           |       20.49       |       22.19      |
> > > > >
> > > > > Table: Evaluation results in accuracy (%). The best performance is highlighted in bold.
> > > > >
> > > > > From the table, we observe that compared with the variant using only top-1 retrieval for augmentation, the inclusion of irrelevant retrieval noise does not significantly degrade the overall performance, demonstrating the robustness of our RoRA-VLM to the noises in retrieval. Furthermore, when we include two additional potentially query-relevant knowledge snippets (as in the top-3 retrieval variant), our RoRA-VLM effectively distinguishes and benefits from the relevant knowledge, resulting in improved performance.
> > > > >
> > > > > ## **All the details and experiments provided in this rebuttal are critical and should be included in the revised version of the paper**
> > > > > We sincerely thank the reviewer for their thoughtful suggestions and comments. We apologize for the omission of these details and experiments in the current draft. We assure the reviewer that all the details and experiments provided in this rebuttal will be incorporated into the revised version of the paper. We will ensure that these additions are appropriately highlighted in BLUE and, if necessary, include them in the appendix with clear references in the main text. We are currently working on the revisions and will post the updated version upon completion.
> > > > >
> > > > > We hope our responses have sufficiently addressed the reviewer’s comments and concerns. If there are any remaining questions or points that require further clarification, please let us know. We kindly request a reevaluation of our work based on the additional experiments and details provided during the rebuttal period. Once again, we sincerely thank the reviewer for their valuable insights, which have been really helpful in improving the clarity, quality, and overall presentation of our manuscript.

---

> > > > > > ### Author Response · Authors · 2024-11-27
> > > > > >
> > > > > > Dear Reviewer XFpf,
> > > > > >
> > > > > > We have uploaded the revised version of our paper based on your comments and suggestions. The updates include all the additional details and experiments we mentioned during the rebuttal period, which are highlighted in blue. We kindly hope you can reevaluate our paper based on the revision.
> > > > > >
> > > > > > Please let us know if you have any further questions or require additional clarification. We sincerely appreciate your time and effort in reviewing our work.
> > > > > >
> > > > > > Best regards,
> > > > > >
> > > > > > Authors

---

> > > > > > > ### Comment · Reviewer_XFpf · 2024-11-27
> > > > > > >
> > > > > > > I thank the authors for clarifying my concerns. Following the detailed rebuttal, I raised my score.

---

### Author Response · Authors · 2024-12-04
**General Response**

Dear Area Chair and Reviewers,

We sincerely thank you for your thoughtful reviews and engagement in the discussion of our paper. We appreciate the insightful comments and recognition of contributions of our work, such as Reviewers XFpf and ZXbA have highlighted that our paper is well-written and emphasizes the necessity and significance of our contributions, noting their potential impact on both academic research and practical applications. All reviewers have acknowledged that our work's motivation is clear and well-articulated.

We are particularly thankful that Reviewer XFpf has increased their score to 6 following our detailed responses and clarifications, and that Reviewer ZXbA has maintained their positive assessment. While Reviewer MWJ4 may not have had the opportunity to engage in further discussions, we believe our responses have effectively addressed their concerns through comprehensive explanations and additional experiments.

As the discussion period concludes, we would like to summarize the key improvements made during the rebuttal phase:

- In response to concerns about baseline comparisons raised by Reviewers XFpf, MWJ4, and ZXbA, we have implemented additional retrieval-augmented vision-language model (VLM) baselines and provided detailed comparisons with existing methods. These comparisons demonstrate the novelty of our contribution as the first work focusing on enhancing the robustness of retrieval-augmented VLMs.

- Following Reviewer XFpf's suggestion, we conducted an extensive analysis of model performance under varying levels of retrieval noise. The results demonstrate our model's effectiveness in mitigating noise in retrieved knowledge, addressing a critical challenge in retrieval-augmented generation.

- In response to feedback from Reviewers MWJ4 and ZXbA, we performed comprehensive ablation studies on our two-stage retrieval process. The results validate its superiority in entity-centric, knowledge-intensive, and complex reasoning tasks.

We would like to emphasize the following contributions, which have been acknowledged by the reviewers:

- Novel Two-Stage Retrieval: Our approach introduces a flexible and modular retrieval-based solution to complex VQA tasks, overcoming the unified multimodal encoding challenges of single-stage retrieval while maintaining robust performance across varying query perspectives.

- Robust Retrieval-Augmented Generation: RORA-VLM effectively addresses the challenge of managing retrieval noise through an innovative combination of visual token refinement and adversarial noise injection, significantly improving performance on knowledge-intensive tasks.

- Effectiveness and Generalizability: Our comprehensive experiments across multiple benchmarks demonstrate substantial improvements over existing approaches, validating our method's effectiveness and practical applicability.

We deeply appreciate the constructive feedback provided by all reviewers. In response, we have carefully refined our work and incorporated all suggested improvements in the revised submission, with updates clearly marked in blue. These changes significantly enhance the clarity and completeness of our paper. We thank the reviewers and area chair for their time and valuable input in helping us improve this work.

Sincerely,

The Authors

---

### Meta-Review · Area_Chair_FFBp · 2024-12-12

**Metareview:**

The paper proposes a retrieval-augmented method for knowledge-intensive tasks to make more relevant use of visual information. The reviewers praise the extensive experiments. However they raise numerous concerns about the method and experiments; some raise a concern about novelty. After the rebuttal stage, there is no strong support for acceptance (all scores are borderline).

**Additional Comments On Reviewer Discussion:**

Two of three reviewers engaged in the discussion; those had borderline accept scores. One, with initially borderline reject, did not participate. These scores are all too close to borderline to provide a strong signal for acceptance.

---

### Decision · Program_Chairs · 2025-01-22

Reject